# MCIF Multimodal Crosslingual Instruction-Following Benchmark from Scientific Talks

**Sara Papi**●, **Maike Züfle**●, **Marco Gaido**●, **Beatrice Savoldi**●, **Danni Liu**●,
**Ioannis Douros**●, **Luisa Bentivogli**●, **Jan Niehues**●

●Fondazione Bruno Kessler (Italy)
●Karlsruhe Institute of Technology (Germany)
●Translated (Italy)

{spapi,mgaido,bsavoldi,bentivo}@fbk.eu
{maike.zuefle,danni.liu,jan.niehues}@kit.edu

## Abstract

Recent advances in large language models have laid the foundation for multimodal LLMs (MLLMs), which unify text, speech, and vision within a single framework. As these models are rapidly evolving toward general-purpose instruction following across diverse and complex tasks, a key frontier is evaluating their crosslingual and multimodal capabilities over both short- and long-form inputs. However, existing benchmarks fall short in evaluating these dimensions jointly: they are often limited to English, mostly focus on a single modality at a time, rely on short-form inputs, or lack human annotations–hindering comprehensive assessment of model performance across languages, modalities, and task complexity. To address these gaps, we introduce MCIF (Multimodal Crosslingual Instruction Following), the first crosslingual human-annotated benchmark based on scientific talks on NLP and beyond. MCIF evaluates instruction following in crosslingual, multimodal settings over different input lengths and spans four macro-tasks: recognition, translation, question answering, and summarization. It covers three core modalities (speech, vision, and text) and four diverse languages (English, German, Italian, and Chinese), fully aligned across all dimensions. This parallel design enables a systematic evaluation of MLLMs' abilities to interpret instructions across languages and effectively integrate multimodal contextual information. Our benchmarking and analysis of 23 models highlight universal challenges across modalities and tasks, indicating substantial room for improvement in future MLLMs development. MCIF is released on 🤗 HuggingFace under CC-BY 4.0 license to promote open research.

## 1 Introduction

In recent years, large language models (LLMs) have achieved remarkable progress across a wide range of tasks (Brown et al., 2020; Grattafiori et al., 2024), leading to a growing interest in extending their capabilities beyond text to embrace multiple modalities such as speech (Rubenstein et al., 2023; Chu et al., 2023; Gaido et al., 2024) and vision (Alayrac et al., 2022; Achiam et al., 2023; Huang et al., 2023). Early efforts typically extended LLMs with a single additional modality and for task-specific applications (Li et al., 2023b; Tang et al., 2023). Building on these foundations, multimodal LLMs (MLLMs) have emerged to unify language, audio, and visual understanding within a single framework (Liang et al., 2024) and are now rapidly evolving toward more flexible and generalized usage, where they are expected to follow natural language instructions and perform diverse, complex tasks (Hendrycks et al., 2020). This paradigm, widely known as *instruction following* (IF), requires models to interpret a user instruction within the provided context and generate an appropriate response across one or more input modalities (Su et al., 2023; Caffagni et al., 2024).

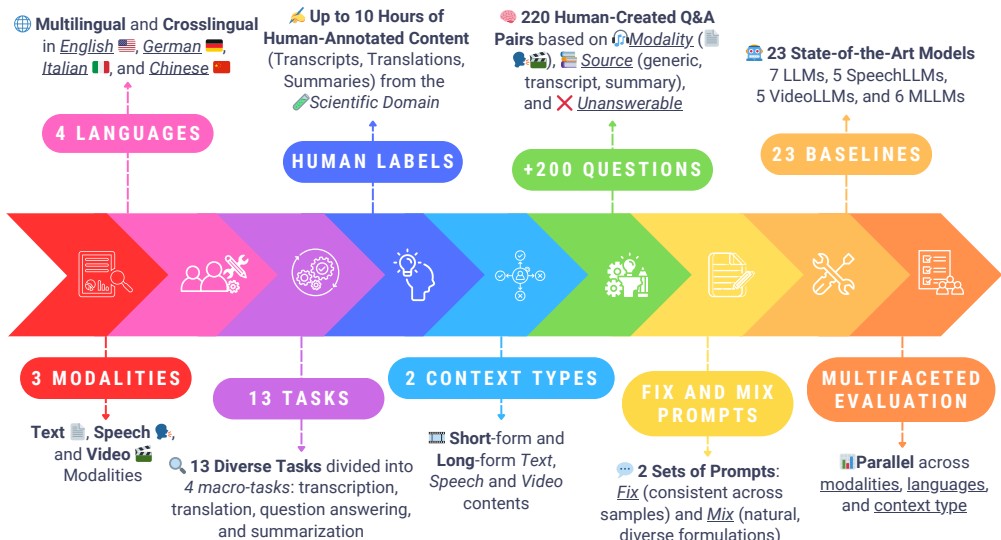

A parallel and equally important challenge lies in extending these capabilities to multilingual and crosslingual settings (Hu et al., 2020; Xu et al., 2025b). General-purpose MLLMs must not only handle inputs and outputs in the same language by supporting the highest possible number of diverse languages, but also process crosslingual multimodal inputs, such as speech in a language paired with an instruction in another language (Zeng et al., 2025). Despite rapid advances in both instruction-following and multilingual modeling (Wei et al., 2021; Sanh et al., 2021; Barrault et al., 2023), current benchmarks fall short of comprehensively analyzing these aspects. Recent work focuses exclusively on two modalities, such as vision-text (Fu et al., 2023; Zhang et al., 2023a; Qian et al., 2024; Das et al., 2024) or speech-text (Yan et al., 2025; Pandey et al., 2025), or restricts its scope to English (Li et al., 2023a; Chen et al., 2024a), thereby overlooking the complexities of multilingual and crosslingual interactions (Duan et al., 2021; Wang et al., 2022; Gao et al., 2023; Pernes et al., 2024). Adding to these limitations, current multimodal benchmarks predominantly focus on short-form inputs, neglecting the evaluation of models' capabilities with long dependencies (Fu et al., 2024), and only a few are human-generated (Li et al., 2023a), raising concerns about data quality, potential biases, and the overall reliability of model evaluations (Zhang et al., 2024a).

To fill this gap, we introduce MCIF,[1] the first manually-curated benchmark explicitly designed to evaluate crosslingual, multimodal IF abilities over both short- and long-form inputs. MCIF covers three core modalities–speech, video, and text–and spans four typologically diverse languages: English, German, Italian, and Chinese. Supporting 4 macro-tasks (recognition, translation, question answering, and summarization), MCIF is fully parallel across modalities and languages, enabling systematic evaluation and ablation studies of MLLMs' abilities to follow instructions across these different dimensions. Our extensive benchmarking and analysis of 23 different systems highlight that, despite recent progress, current models still face significant challenges: they struggle to handle long-form contexts–especially when tasked to summarize their content, to jointly integrate speech and video effectively, and to answer fine-grained, content-specific questions. These findings highlight main directions for improving crosslingual and multimodal processing in IF systems.

## 2 RELATED WORKS

In this section, we survey existing IF benchmarks for speech and vision, highlighting critical gaps in crosslingual, multimodal, and long-form evaluation that MCIF is designed to address.

**Speech-Text IF Benchmarks.** Most existing efforts in IF speech-text evaluation datasets, such as Speech-ifeval (Lu et al., 2025), SAKURA (Yang et al., 2025b), and MMSU (Wang et al., 2025b),

---

[1]The benchmark is released under CC-BY 4.0 license at hf.co/datasets/FBK-MT/MCIF to facilitate research and broad adoption. The inference and evaluation code are available under Apache 2.0 license at github.com/hlt-mt/mcif, which also contains the systems' outputs under CC-BY 4.0 license.

| task name | acronym | in mod | out mod | src lang | tgt lang | context type | cross |
|---|---|---|---|---|---|---|---|
| Textual Question Answering | TQA | 📄 | 📄 | 🇺🇸 | 🇺🇸🇩🇪🇮🇹🇨🇳 | LONG | ✔ |
| Text Summarization | TSUM | 📄 | 📄 | 🇺🇸 | 🇺🇸🇩🇪🇮🇹🇨🇳 | LONG | ✔ |
| Machine Translation | MT | 📄 | 📄 | 🇺🇸 | 🇩🇪🇮🇹🇨🇳 | LONG | ✔ |
| Automatic Speech Recognition | ASR | 🗣 | 📄 | 🇺🇸 | 🇺🇸 | SHORT LONG | ✘ |
| Spoken Question Answering | SQA | 🗣 | 📄 | 🇺🇸 | 🇺🇸🇩🇪🇮🇹🇨🇳 | SHORT LONG | ✔ |
| Speech Summarization | SSUM | 🗣 | 📄 | 🇺🇸 | 🇺🇸🇩🇪🇮🇹🇨🇳 | LONG | ✔ |
| Speech Translation | ST | 🗣 | 📄 | 🇺🇸 | 🇩🇪🇮🇹🇨🇳 | SHORT LONG | ✔ |
| Video Question Answering | VQA | 🎞 | 📄 | 🇺🇸 | 🇺🇸🇩🇪🇮🇹🇨🇳 | SHORT LONG | ✔ |
| Video Summarization | VSUM | 🎞 | 📄 | 🇺🇸 | 🇺🇸🇩🇪🇮🇹🇨🇳 | LONG | ✔ |
| Audio-Video Recognition | AVR | 🎞🗣 | 📄 | 🇺🇸 | 🇺🇸 | SHORT LONG | ✘ |
| Audio-Video Question Answering | AVQA | 🎞🗣 | 📄 | 🇺🇸 | 🇺🇸🇩🇪🇮🇹🇨🇳 | SHORT LONG | ✔ |
| Audio-Video Summarization | AVSUM | 🎞🗣 | 📄 | 🇺🇸 | 🇺🇸🇩🇪🇮🇹🇨🇳 | LONG | ✔ |
| Audio-Video Translation | AVT | 🎞🗣 | 📄 | 🇺🇸 | 🇩🇪🇮🇹🇨🇳 | SHORT LONG | ✔ |

Table 1: Tasks in MCIF with their input/output modalities (**in mod**, **out mod**), input type (**context type**) that can be long-form ( LONG ) or short-form ( SHORT ), source/target languages (**src lang**, **tgt lang**) among English 🇺🇸, German 🇩🇪, Italian 🇮🇹, and Chinese 🇨🇳. Since all IF tasks involve text prompts, we report 📄 when a task uses only the text modality. **cross** indicates whether the task can be crosslingual, i.e., if it involves a target language different from the source language. The detailed description of each task is provided in Appendix A.

restrict their scope to instruction-following monolingual tasks, predominantly covering the English language. AIR-Bench (Yang et al., 2024), VoiceBench (Chen et al., 2024a), ADU-Bench (Gao et al., 2024), URO (Yan et al., 2025), and SpeechInstructBench (Wang et al., 2025c) are more dialogue-oriented tasks that are limited to English and Chinese, with the latter three benchmarks relying entirely on synthetic speech. SD-Eval (Ao et al., 2024), Dynamic-SUPERB (Huang et al., 2025), AudioBench (Wang et al., 2025a), MSTEB (Beyene et al., 2025), and SIFT-50M (Pandey et al., 2025) offer a multilingual speech-text evaluation but rely on preexisting benchmarks, such as CommonVoice (Ardila et al., 2020), and FLEURS (Conneau et al., 2023), making them prone to data contamination (Sainz et al., 2023; Balloccu et al., 2024; Kocyigit et al., 2025) and limited to short-form, speech-only assessment. Overall, while interest in speech-text evaluation is growing, existing benchmarks do not support multimodal, crosslingual, and long-form instruction-following in a unified setting as MCIF.

**Vision-Text IF Benchmarks.** Similar to the speech-text domain, the vision-text domain has seen a huge increase in the number of benchmarks designed to assess MLLMs across diverse capabilities. MMMU (Yue et al., 2024) and MIA-Bench (Qian et al., 2024) evaluate MLLMs with image-textual inputs across several domains, but cover English only. MME (Fu et al., 2023) extends the evaluation to Chinese-to-English translation, and M3Exam (Zhang et al., 2023a) to 9 diverse languages, while EXAMS-V (Das et al., 2024) further widens the coverage to 11 languages. Despite their extensive language coverage, these vision-text benchmarks are all constrained to benchmark models' abilities when dealing with single images rather than videos–sequences of images. Video-based benchmarks such as Video-Bench (Ning et al., 2023), InfiniBench (Ataallah et al., 2024), VITATECS (Li et al., 2024b), TempCompass (Liu et al., 2024), LVBench (Wang et al., 2024b), MVBench (Li et al., 2024a), and MMBench-Video (Fang et al., 2024) focus on bimodal interactions (video and text), cover English only, and rarely incorporate human-authored multilingual instructions.

VideoMME (Fu et al., 2024) and MF$^2$ (Zaranis et al., 2025) are the first benchmarks comprising the three modalities (speech, text, and video); however, VideoMME is not crosslingual and restricts its scope solely to video-centric tasks, while MF$^2$ includes speech but does not evaluate this modality. As a result, no benchmark currently enables systematic evaluation across speech, video, and text modalities in a crosslingual instruction-following framework.

# 3 MULTIMODAL CROSSLINGUAL INSTRUCTION-FOLLOWING BENCHMARK

We create MCIF from English videos of scientific presentations (from NLP topics and beyond), including their related audio, by manually creating transcripts and translations of their content, summaries (abstracts), and a set of questions and open-ended answers. It results in a highly multi-task, natural, human-labeled, and expert-vetted benchmark characterized by: *i)* **3 modalities**: text, speech, and video; *ii)* **4 languages**: English, German, Italian, and Chinese; *iii)* **2 context types**: short-form and long-form text, speech, and video contents; and *iv)* **13 tasks**: crosslingual and multimodal tasks, which are divided into 4 macro-tasks (recognition, translation, question answering, and summarization), and reported in Table 1. Each sample is composed of the *input content* (either short- or long-form text, speech, or video), which is paired with a *textual prompt* containing the instruction to be followed (in English, German, Italian, or Chinese), and its corresponding *textual reference* (transcription, translation, summary, or answer in the same language as the prompt). MCIF is designed to be parallel across languages and modalities, as each sample contains the input in all three modalities, and prompts and outputs in all four languages. We describe the data selection in Section 3.1, the human-annotation process in Section 3.2, and the instruction-following prompt composition in Section 3.3.

## 3.1 DATA SELECTION AND COLLECTION

We collected scientific talks from the ACL Anthology, the main reference repository for research in the language technologies community. This source is well-suited to our objective since *i)* it is openly available under a *CC-BY 4.0 License*, allowing unrestricted use and redistribution; *ii)* it offers naturally multimodal and challenging material, i.e., video presentations self-recorded by speakers from various linguistic backgrounds and accents, accompanied by slides, spoken audio, and corresponding research papers. To avoid data contamination issues of testing models on material that has been used for training (Balloccu et al., 2024; Kocyigit et al., 2025), we selected the most recent available material at the time of collection,[2] namely, the ACL 2023 paper presentations.

We randomly picked videos from the ACL 2023 main conference papers, covering different topics in the context of NLP and beyond, spanning from multimodality to explainability and ethics. The resulting selection is both diverse in terms of recording conditions (each presenter recorded their talk independently, using their own equipment and environment, with significant variation in audio and video quality, background, and presentation style), and speaker demographic, as ACL is an international conference that features researchers from around the world (representing a wide range of nationalities from Europe, the Americas, Asia, and beyond).[3] The collection was manually inspected and validated to discard presentations with *i)* repeated speakers (i.e., each sample represents a unique speaker), *ii)* inaudible or low-quality speech (e.g., presentations with excessive background noise or featuring a speaker distant from the microphone), and *iii)* automatically generated speech (e.g., text-to-speech synthesis is used to produce the audio). The resulting benchmark includes 21 presentations, with a total of 2 hours of video content and approximately 15.5k words. The videos are released in their original `mp4` format, and they are converted into mono-channel, 16 kHz `wav` audios. To support both the exploration of how models handle *long* versus *short* context and to maximize usability for models with limited context capacity, we provide both the full-length video/speech context and an automatically segmented version generated with SHAS (Tsiamas et al., 2022), with segments of ∼16 seconds. Together with the video, we collect the abstracts, which serve as summaries for the presentations. To improve test set representativeness for summarization, we further collect 79 additional videos, yielding a total of 100 samples–about 10 hours of content with summaries totaling ∼17k words. For these additional samples, audio and textual transcripts are also available, ensuring alignment across all three modalities. A detailed breakdown of MCIF statistics is provided in Fig. 1.

## 3.2 DATASET MANUAL ANNOTATIONS

We describe the MCIF curation process, including the human annotation. Key steps are summarized below; details on costs, annotation, and design guidelines are in Appendix B.

---

[2] As of April 18[th], 2025, the most recent conference with available videos is chosen for the benchmark.

[3] https://aclanthology.org/2023.acl-long.report.pdf

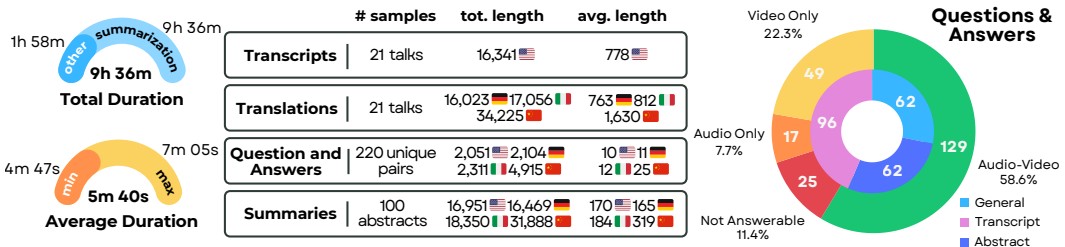

Figure 1: Breakdown of MCIF statistics. Total length is measured in space-separated words for English, German, and Italian, and in characters for Chinese. Question-answer statistics in the inner circle refer to the question type, while the outer circle refers to the input modality (see Section 3.2).

**Recognition and Summarization.** For each talk, we tasked professional linguists with producing high-quality gold transcripts in US English, following detailed guidelines (see Appendix B). This enabled the creation of aligned English video, speech, and text data. For summarization, instead, we used the abstract from the associated paper as the textual summary, as prior work shows abstracts provide reasonably accurate representations of the scientific talks (Züfle et al., 2025).

**Question-Answering.** To evaluate model understanding, we design a QA task intended to probe different aspects of contextual dependency and task realism. First, each talk was paired with at least 10 QA pairs, which followed a structured distribution: *i) General common questions*, which are generic and applicable to any talk (e.g., "What are the affiliations of the authors?"); *ii) Transcript questions*, created after watching the full talk and targeting narrow, context-dependent information retrieval; and *iii) Abstract questions*, generated after reading only the abstract, simulating a scenario where a user queries about a talk without having watched it in its full length. All QA pairs were created and verified by 16 expert annotators with high English proficiency and a background in machine learning and NLP. Each QA pair was also annotated with the input modality required to answer the question, explicitly including cases where no answer is available. Labels were assigned as follows: **NA** if the information was not present in either the video or audio, **AV** if the information was explicitly available in both the audio and video modalities, with either modality alone being sufficient to answer the question, **A** if the answer was explicit in the audio only, and **V** if it was explicit in the video only. A breakdown of the QA distribution among categories is illustrated in Fig. 1. Overall, this setup enables a systematic evaluation of model performance across modality conditions and unanswerable cases, as detailed in Appendix B. All QA pairs are created in English and, as described in the following, are then translated into three additional languages.

**Translation and Crosslinguality.** To make MCIF crosslingual, all English textual data–transcripts, summaries, and QA pairs–were translated into three additional languages: Italian, German, and Chinese (Mandarin). These languages were selected as they are well-resourced, allow for comparable analyses, and represent a diversity of language (sub)families and writing systems. All translations were carried out by professional translators with expertise in scientific content. As translators reviewed the original QA pairs, summaries, and transcripts during this process, translation also served as a secondary verification step, further ensuring the quality and consistency of the source material.

## 3.3 INSTRUCTION-FOLLOWING PROMPTS

For each instance in the benchmark, information such as the specific task (e.g., ASR), the input modality (e.g., audio, video, or text), or the target language (e.g., German) is not provided as explicit metadata; rather, the model must infer these aspects from the prompt itself, simulating real human interaction, and fulfill diverse instructions across the supported language pairs (e.g., "Rispondi in modo conciso alla seguente domanda dato il contenuto inglese: {QUESTION}"[it], "Answer the following question concisely given the English content: {QUESTION}"[en]). Following previous work, we always specify the source language in the prompt, which is written in the target language (Xu et al., 2024; Lu et al., 2024a).

We create two variants of the MCIF benchmark, MCIF$_{fix}$ and MCIF$_{mix}$, based on the set of prompts. MCIF$_{fix}$ employs a fixed prompt for each macro-task (recognition, translation, question answering, and summarization). Instead, MCIF$_{mix}$ selects a prompt at random from a pool of ten alternatives

for each macro-task, where the pool includes the fixed prompt from $MCIF_{fix}$. By contrasting the two settings–always using the same prompt versus sampling from diverse ones–we can directly measure the generalization and robustness of models to different prompt wordings. All prompts were manually crafted for each of the four languages and are reported in Appendix C.

# 4 EXPERIMENTAL SETTINGS

**Models.** We evaluate a range of models across modalities: LLMs on textual tasks, SpeechLLMs on speech tasks, VideoLLMs on video-only tasks (without speech), and MLLMs on all tasks (text, speech, video, and speech+video). To ensure compatibility within a unified evaluation framework, we select publicly available state-of-the-art open-weight models hosted on HuggingFace that can be run using the HuggingFace Transformers library. Due to computational constraints, we restrict our selection to models with fewer than 20 billion parameters. Additionally, we evaluate a commercial MLLM, Gemini 2.5 Flash (Comanici et al., 2025), whose outputs are obtained through API calls. This results in 23 models: 7 LLMs, 5 SpeechLLMs, 5 VideoLLMs, and 6 MLLMs. The full models list, and generation settings are detailed in Appendix D.

**Metrics.** The evaluation is carried out by computing separate scores for each of the tasks addressed, using commonly adopted metrics in the community. Namely, for recognition tasks (ASR, AVR), we compute WER using the jiWER library after normalizing the test using the Whisper normalizer (Radford et al., 2023), version `0.0.10`. For translation tasks (MT, ST, AVT), we use COMET (Rei et al., 2022), with the standard model `Unbabel/wmt22-comet-da`, after concatenating all (speech or video) segments belonging to the same talk in the case of the short context and resegmenting the text with `mwerSegmenter` (Matusov et al., 2005) to pair them with the reference sentences. Lastly, for question answering (TQA, SQA, VQA, AVQA) and summarization (TSUM, SSUM, VSUM, AVSUM), we compute BERTScore (Zhang et al., 2020) rescaled with the baseline to make scores more interpretable, with 0 corresponding to random outputs in the target language.

# 5 RESULTS

This section reports results on MCIF from several perspectives: Section 5.1 analyzes overall performance across all 23 models and macro-tasks, Section 5.2 focuses on MLLMs to study how different modalities impact task performance, and Section 5.3 studies how the best model in each category (LLM, SpeechLLM, VideoLLM, MLLM) performs on question answering across question types.

## 5.1 OVERALL RESULTS

Table 2 reports the model results on $MCIF_{fix}$ and $MCIF_{mix}$ for each context (long or short), and macro-task. Extended results per language are reported in Appendix E.

**RECOGNITION** Some SpeechLLMs (Phi4-Multimodal, GraniteSpeech) and MLLMs (Ola, Gemini 2.5 Flash) show strong performance (WER<10), demonstrating the feasibility of this macro-task. However, despite being the simplest one–and on which all models are trained on–several systems fail in one or both context types. UltraVox v0.5, Ming-Lite-Omni, and MiniCPM-o-2 achieve scores superior to 100 WER on both short- and long-form, DeSTA2 exceeds 100 WER on long-form, and, surprisingly, Ola drops sharply from 6.6/14.0 (long-form) to 98.8/104.1 (short-form). A manual inspection of the model's outputs revealed that Ola often misinterprets transcription prompts in short-form, opting instead to perform image captioning of accompanying slides, while this is not the case for long-form inputs (see Example 1 in Appendix F). The long-form results of Ola also suggest that its architecture–particularly the strategy of chunking and concatenating long speech segments using a Whisper-based encoder–enables the model to obtain high transcription quality over extended inputs.

**TRANSLATION** As expected, LLMs dominate due to the maturity of text-based translation, a trend consistent across target languages (see Appendix E). Beyond LLMs, some SpeechLLMs and MLLMs achieve competitive results in short-form, with COMET >75 for DeSTA2, Ola, and Qwen2.5-Omni, and even >80 for Phi4-Multimodal. However, several models fail to perform this task, either across all conditions (UltraVox v0.5, MiniCPM-o-2) or on long-

| Context | Input Modality | Model | MCIF$_{fix}$ | | | | MCIF$_{mix}$ | | | |
|---|---|---|---|---|---|---|---|---|---|---|
| | | | **REC** WER↓ | **TRANS** COMET↑ | **QA** BERTS.↑ | **SUM** BERTS.↑ | **REC** WER↓ | **TRANS** COMET↑ | **QA** BERTS.↑ | **SUM** BERTS.↑ |
| SHORT | 🗣️ | DeSTA2 | 54.0 | 75.3 | 17.2 | × | 83.0 | 75.2 | 18.6 | × |
| | | GraniteSpeech | 9.4 | 52.1 | 0.5 | | 9.5 | 46.6 | 0.4 | |
| | | Phi4-Multimodal | **6.8** | **80.2** | 37.1 | | **6.7** | 80.1 | 37.4 | |
| | | Qwen2-Audio | 31.7 | 74.9 | 32.6 | | 31.9 | 74.6 | 32.8 | |
| | | UltraVox v0.5 | 127.7 | 43.3 | 19.6 | | 172.6 | 43.2 | 19.1 | |
| | 🎞️ | InternVL3 | × | × | 31.7 | × | × | × | 31.3 | × |
| | | LLaVA-NeXT | | | 13.7 | | | | 12.1 | |
| | | Qwen2.5-VL | | | 39.1 | | | | 37.8 | |
| | | VideoLLaMA3 | | | 24.1 | | | | 23.8 | |
| | | Video-XL2 | | | 13.6 | | | | 13.6 | |
| | 🗣️🎞️ | Gemma 3n | 35.1 | 73.0 | 26.2 | × | 58.9 | 71.5 | 25.1 | × |
| | | Ming-Lite-Omni | 117.5 | 53.0 | 15.8 | | 128.2 | 53.3 | 13.3 | |
| | | MiniCPM-o-2 | 144.8 | 39.7 | 21.4 | | 207.1 | 38.8 | 23.1 | |
| | | Ola | 104.1 | 76.6 | 37.3 | | 98.8 | 76.3 | 37.0 | |
| | | Qwen2.5-Omni | 43.5 | 77.3 | 34.3 | | 48.0 | 76.5 | 35.1 | |
| | | *Gemini 2.5 Flash* | 14.9 | 67.0 | **40.6** | | 12.8 | 69.2 | **39.5** | |
| LONG | 📄 | Aya Expanse | × | 68.7 | 26.7 | **25.1** | × | 68.7 | 23.1 | 24.2 |
| | | Gemma 3 | | 85.5 | 22.9 | 9.5 | | 83.4 | 21.8 | 9.4 |
| | | GPT-oss | | 75.0 | 24.6 | 17.8 | | 70.1 | 20.9 | 15.9 |
| | | Llama 3.1 | | 81.4 | 30.3 | 24.8 | | 79.5 | 31.0 | **24.5** |
| | | Phi4 | | 84.5 | 30.8 | 13.0 | | **84.7** | 29.6 | 14.5 |
| | | Qwen3 | | 84.8 | 37.9 | 19.9 | | 84.5 | 35.6 | 20.1 |
| | | Tower+ | | **85.6** | 29.5 | 19.8 | | 83.7 | 23.4 | 17.6 |
| | 🗣️ | DeSTA2 | 112.9 | 41.3 | 12.5 | 3.0 | 132.5 | 40.8 | 12.6 | 2.8 |
| | | GraniteSpeech | 99.9 | 36.0 | -23.7 | 2.8 | 80.4 | 34.6 | -22.8 | -10.5 |
| | | Phi4-Multimodal | 39.2 | 59.7 | 37.6 | 7.4 | 29.8 | 59.5 | 37.3 | 17.9 |
| | | Qwen2-Audio | 92.9 | 41.0 | 28.8 | 7.3 | 93.1 | 41.1 | 28.9 | 6.2 |
| | | UltraVox v0.5 | 89.1 | 38.0 | 12.7 | -3.0 | 92.5 | 38.0 | 12.5 | -3.1 |
| | 🎞️ | InternVL3 | × | × | 27.6 | 20.4 | × | × | 27.9 | 20.4 |
| | | LLaVA-NeXT | | | 7.2 | -7.0 | | | 5.2 | -6.7 |
| | | Qwen2.5-VL | | | 33.7 | 23.3 | | | 34.9 | 20.2 |
| | | VideoLLaMA3 | | | 26.8 | -19.9 | | | 26.5 | -33.0 |
| | | Video-XL2 | | | 17.2 | 3.7 | | | 17.4 | 4.1 |
| | 🗣️🎞️ | MiniCPM-o-2 | 170.6 | 24.9 | -38.0 | -39.1 | 179.2 | 25.9 | -38.4 | -39.7 |
| | | Ola | 14.0 | 63.2 | 36.2 | 12.3 | **6.6** | 58.7 | 36.2 | 13.8 |
| | | Qwen2.5-Omni | 98.5 | 47.5 | 32.5 | 8.9 | 94.9 | 40.2 | 34.8 | 9.4 |
| | | *Gemini 2.5 Flash* | **11.9** | 76.4 | **46.1** | 24.1 | 7.9 | 79.9 | **45.9** | 21.8 |

Table 2: Overall results, averaged across languages, on MCIF$_{fix}$ and MCIF$_{mix}$, divided into the four macro-tasks: recognition (**REC**), translation (**TRANS**), question answering (**QA**), and summarization (**SUM**). Best result by context is marked in **bold**, and best overall result is underlined. × marks unfeasible tasks, i.e., summarization with short-form, or out of models' scope for a given modality: recognition for LLMs and VideoLLMs (requires speech), and translation for VideoLLMs (requires speech or text). Gemma 3n and Ming-Lite-Omni are removed from LONG as they are not able to process long-form inputs.

form (DeSTA2🗣️, GraniteSpeech🗣️, Qwen2-Audio🗣️, Qwen2.5-Omni🗣️🎞️), where scores drop below 50 COMET. The degradation is often due to under-translation, with models skipping parts of the context–especially in long-form (see Example 2 in Appendix F). An exception is Gemini 2.5 Flash🗣️🎞️, which performs better on long-form; a manual inspection revealed that, on shorter segments, it hallucinates or over-elaborates on audio/video content (see Example 3 in Appendix F), a known issue in current LLMs (Briakou et al., 2024).

**QUESTION ANSWERING** Surprisingly, not all LLMs excel in question answering even if provided with human transcripts, as the best performance consistently comes from Gemini 2.5 Flash🗣️🎞️. Results are inconsistent, particularly in short-form, where the top models vary by language (see Appendix E): SpeechLLMs (Phi4-Multimodal), VideoLLMs (Qwen2.5-VL), and MLLMs (Gemini 2.5 Flash). In contrast, Gemini 2.5 Flash🗣️🎞️ clearly dominates in long-form, consistently ranking first across languages with average BERTScores above 45. SpeechLLMs and VideoLLMs, instead, suffer significant performance drops, echoing trends observed in other tasks. Only a few models fail

entirely, such as LLaVA-NeXT⊞ (BERTScore <10, corresponding to outputs in the wrong language; see Example 4 in Appendix F) and GraniteSpeech🎙 (BERTScore around 0, corresponding to random text in the correct language or the transcript of the content; see Example 5 in Appendix F).

**SUMMARIZATION** This is by far the most challenging macro-task, with some systems even producing negative BERTScores–worse than random outputs in the target language. Failures span SpeechLLMs (GraniteSpeech, UltraVox v0.5), VideoLLMs (LLaVA-NeXT, VideoLLaMA3), and MLLMs (MiniCPM-o-2). Manual inspection points to two recurring issues: models either default to the wrong language (often English across tasks; see Example 4 in Appendix F) or ignore the instruction altogether (e.g., LLaVA-NeXT⊞ transcribes slides instead of summarizing them; see Example 1 in Appendix F). LLMs achieve the strongest results, confirming that text-only inputs remain easier to handle, followed by MLLMs, whose performance fluctuates widely (from the negative scores of MiniCPM-o-2 to the good scores of Gemini 2.5 Flash). VideoLLMs exhibit similar instability, although generally at lower BERTScores. SpeechLLMs remain weak, with the sole exception of Phi4-Multimodal🎙.

SHORT ⚔ LONG Across tasks, models generally perform better on short-form inputs, with long-form leading to notable degradation–particularly for SpeechLLMs and VideoLLMs. An exception is Gemini 2.5 Flash🎙⊞–with significant improvements in all tasks–and Ola🎙⊞ in recognition. Despite this, long-form recognition remains a major challenge for most systems, regardless of their input modality. Manual inspection revealed that the main source of degradation is under-generation, with models producing only partial outputs (see Example 2 in Appendix F). This is particularly common in recognition but also in translation: for instance, DeSTA2🎙 and Qwen2-Audio🎙 drop by about 34 COMET, while Qwen2.5-Omni🎙⊞ falls by roughly 30 COMET. In contrast, most MLLMs improve or maintain performance on long-form question answering, notably Ola🎙⊞ and Qwen2.5-Omni🎙⊞. Additional failure cases that are especially pronounced in long-form settings include persistent use of the wrong language (GraniteSpeech🎙 defaulting to English; see Example 4 in Appendix F), common in all macro-tasks, or refusal to answer the user's requests (UltraVox v0.5🎙; see Example 6 in Appendix F). Lastly, some MLLMs (Gemma 3n and Ming-Lite-Omni) cannot handle long-form inputs at all, constrained by limited context windows.

**MCIF_fix ⚔ MCIF_mix.** Comparing the two MCIF variants reveals that most models exhibit limited robustness to prompt reformulation, with sensitivity varying across tasks. Recognition is the most affected: some SpeechLLMs (DeSTA2, and UltraVox v0.5) and MLLM (MiniCPM-o-2) fluctuate up to more than 60 WER on short-form, and nearly all systems vary on long-form, with shifts up to 20 WER (e.g., GraniteSpeech🎙). Translation remains relatively stable for LLMs, even if drops of up to 4.9 COMET occur (e.g., GPT-oss), with SpeechLLMs showing similar patterns. MLLMs prove less reliable, particularly in long-form, with Qwen2.5-Omni🎙⊞ losing up to 7 COMET. Question answering is generally stable, but most LLMs show notable variations, with changes up to 6.1 BERTScore (Tower+). Summarization follows an unclear trend: some models show little robustness to prompt reformulation (GraniteSpeech🎙, Phi4-Multimodal🎙, VideoLLaMA3⊞), with variations up to 13.1 BERTScore, while others remain consistent.

To sum up, results reveal consistent trends in current models' performance. Summarization is the most difficult task, with no gains from adding speech or video to text–underscoring limitations in multimodal integration. QA shows the opposite pattern, benefiting from speech or video and highlighting the value of non-textual modalities. LLMs continue to lead in translation, while recognition proves highly sensitive to prompt variability. Long-form proves challenging, with nearly all models suffering significant drops across tasks compared to short-form, especially SpeechLLMs and MLLMs. Together, these findings expose the wide gap between current systems and the goal of robust, multimodal, crosslingual instruction following, pointing to clear avenues for future progress.

## 5.2 EFFECT OF MODALITIES INTEGRATION

Since MCIF is completely parallel across languages and modalities, it enables an ablation study on how different modalities contribute to MLLM performance. Specifically, we evaluate each model under four input conditions: text only, speech only, video only, and speech+video (as already reported in Table 2). To isolate the contribution of each modality, we use the MCIF_fix set to avoid biases from the single fixed prompt in MCIF_fix that could favor some models over others, and run the evaluation on both short and long contexts. The results are shown in Fig. 2.

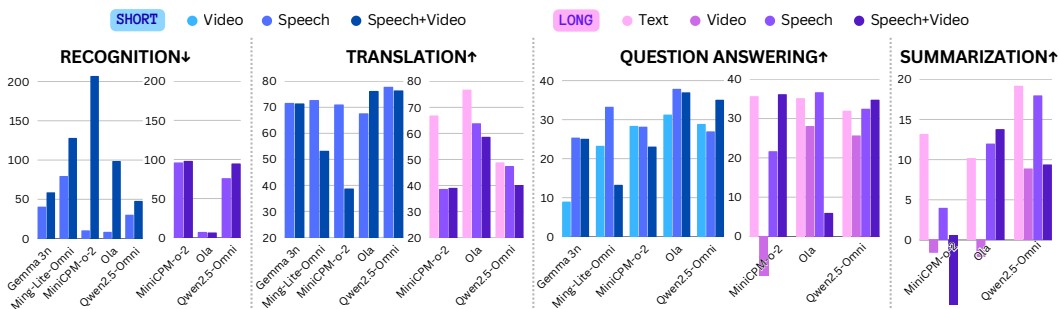

Figure 2: MLLM results on MCIF$_{mix}$ by inference modality, averaged across languages.

In recognition, comparing Speech and Speech+Video in both short- and long-form, we observe that the video modality provides no benefit and often degrades performance when combined with speech–except in one case (Ola on long-form). In translation, speech leads the performance in short-form, with Ola being the only model to benefit from the addition of video (improving by 8.6 COMET), while in long-form the text modality–available only in this setting–dominates, followed by speech. For short-form question answering, speech again proves fundamental, while video consistently underperforms. The only case where video outperforms speech is Qwen2.5-Omni, but the margin is minimal (1.9 BERTScore); notably, it is also the only model where combining speech and video brings a clear gain over speech alone. In long-form question answering, video alone yields negative scores (MiniCPM-o-2), confirming its limited exploitation in current MLLMs, but joint speech-video processing shows benefits in two out of three models (MiniCPM-o-2 and Qwen2.5-Omni). Summarization trends are less consistent across systems: in two out of three models, text enables the best or comparable results, while video-only produces negative scores (MiniCPM-o-2 and Ola), and joint speech-video processing even harms performance in one case (MiniCPM-o-2). Overall, these findings indicate that current MLLMs struggle to effectively integrate speech and video, with joint multimodal processing often providing no benefit or even being counterproductive. Moreover, the video modality–despite showing the good results in short-form question answering (Table 2)–remains underutilized in MLLMs, systematically yielding the weakest results.

## 5.3 BREAKDOWN ON QUESTION ANSWERING

To better understand model behavior beyond overall QA scores, we analyze performance across different question types (Section 3.2). Breaking results down by question modality (Audio-Visual, Audio, and Video) and by source (General, Abstract, Transcript) helps reveal how well models exploit specific input signals and how they handle varying levels of specificity and difficulty. For this analysis, we use long-form contexts, which enable evaluation of all models, including LLMs, and report results on MCIF$_{mix}$ to avoid biases from the single fixed prompt in MCIF$_{fix}$, which could favor some models over others. Scores are averaged across languages, and the best model from each family–Qwen3, Phi4-Multimodal, Qwen2.5-VL, and Ola–is shown in Fig. 3.

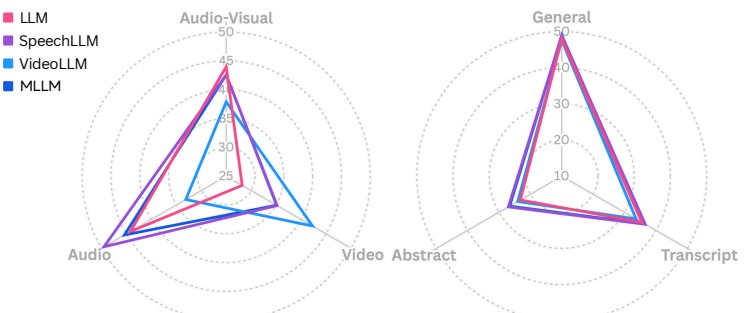

Figure 3: Performance breakdown on MCIF$_{mix}$ LONG QA of the best models by question modality and source.

We find that audio-only (A) questions are best handled by the SpeechLLM, while the VideoLLM performs strongest on video-only (V) questions. Surprisingly, MLLMs underperform even if accessing both modalities. The LLM, despite working only on transcripts, achieves the highest score on audiovisual (AV) questions (44), slightly surpassing both the SpeechLLM and MLLM

(42.6), confirming that text remains easier to process than multimodal inputs. As expected, modality mismatches cause substantial drops: SpeechLLMs underperform on V questions and VideoLLMs on A questions (losses of 7-13 points), though both remain above 35, likely thanks to contextual cues. These results reveal that, despite being explicitly designed for multimodality, MLLMs still fail to effectively integrate speech and visual signals together, leaving substantial room for improvement. Breaking results down by question source reveals a consistent trend: generic questions (General) yield the highest scores (47.6-49.0), while *talk-specific* ones prove more challenging as performance drops to 33.7-36.6 on Transcript questions and further to 23.2-27.0 on Abstracts. This suggests that current models excel at retrieving common information (e.g., paper title or affiliations) but remain weak at retrieving fine-grained content, regardless of modality.

# 6  CONCLUSIONS

In this work, we introduced MCIF, the first human-annotated multimodal and crosslingual instruction-following benchmark from the scientific domain. MCIF spans three modalities (text, speech, and video) and four typologically diverse languages (English, German, Italian, and Chinese), parallel across all dimensions. It incorporates both short- and long-form contexts and covers 13 tasks, organized into four macro-tasks: recognition, translation, question answering, and summarization. MCIF comprises two variants–$\text{MCIF}_{\text{fix}}$ with fixed prompts and $\text{MCIF}_{\text{mix}}$ with diverse ones–whose comparison assesses models' robustness and generalization to controlled instruction reformulation under preserved semantic equivalence.

Through extensive benchmarking of 23 state-of-the-art LLMs, SpeechLLMs, VideoLLMs, and MLLMs, we identified both their strengths and significant limitations, particularly regarding joint speech and video modality integration, long-form processing, and summarization. These findings point to key directions for future system design: improving multimodal fusion to process multiple modalities jointly and enhancing robustness to long-form content via sequence compression or extended context modeling. In addition, our results on $\text{MCIF}_{\text{fix}}$ vs. $\text{MCIF}_{\text{mix}}$ further show that robustness to prompt variation remains an open challenge. While our evaluation focuses on single-turn, simple instructions–reflecting common practice and training paradigms of current multimodal models–future work can leverage MCIF as a foundation to explore more complex prompting strategies, including prompt engineering (Liu et al., 2023; Chen et al., 2025), prompt embedding alignment (Kim & Angelova, 2025), contrastive training between equivalent instructions (Yan et al., 2024), instruction tuning based on carefully selected data (Qin et al., 2025), preference optimization (Ouyang et al., 2022), and chain-of-thought reasoning (Wei et al., 2022; Zhang et al., 2023b).

Overall, MCIF provides a comprehensive evaluation framework and establishes a foundation for advancing general-purpose, multimodal, and crosslingual instruction-following systems.

ACKNOWLEDGMENTS

The work presented in this paper is funded by the European Union's Horizon research and innovation programme under grant agreement No 101135798, project Meetween (My Personal AI Mediator for Virtual MEETings BetWEEN People), and the PNRR project FAIR - Future AI Research (PE00000013), under the NRRP MUR program funded by the NextGenerationEU.

## ETHIC STATEMENT

**Licensing and Attribution.** The MCIF benchmark is derived from publicly available data released under a CC-BY 4.0 license, and we release it under the same terms to support transparent and open science. As described in Section 3.2, all manual annotations were performed by the authors, colleagues, or compensated professionals, with informed consent; details of compensation are given in Appendix B. Contributors will be acknowledged in the non-anonymous version of this work.

**Use of Flags for Languages.** In the paper, we use flags to denote the languages represented in MCIF. We acknowledge that this practice raises ethical and sociolinguistic concerns, since flags symbolize countries or national entities rather than languages, and may be misleading in multilingual contexts. In our dataset, however, each flag corresponds to a specific language variant used in MCIF

transcripts and translations (e.g., US English for *en*, German for *de*, Italian for *it*, Mandarin for *zh*), which makes this representation informative for our use case.

**Use of Large Language Models.** For the writing process, ChatGPT was employed exclusively to correct grammar in content authored by humans.

## REPRODUCIBILITY STATEMENT

We provide a detailed description of the collection process for our newly introduced dataset MCIF in Section 3.1, and report comprehensive information on the manual annotations in Section 3.2. The annotation process is detailed in Appendix B, including information on annotator recruitment through a professional agency, compensation, the number of annotators involved, and the tools employed during the annotation. For transparency, we release the complete annotation guidelines: the instructions for the transcription and translation tasks, and the guidelines for the question answering task are available at `https://github.com/hlt-mt/mcif/tree/main/dataset_build/annotation_guidelines`, as also referenced in Appendix B. For the MCIF baselines, we provide the full list of models used in Appendix D, including model references, links to the HuggingFace model weights, generation settings, and corresponding transformer versions. The prompts are part of our dataset and are listed in Appendix C. We release all code for reproducing the baselines, along with the evaluation scripts, at `https://github.com/hlt-mt/mcif`. In addition, we release the outputs generated by each model with the same license as the benchmark, namely CC-BY 4.0, at `https://github.com/hlt-mt/mcif/tree/main/baselines/outputs`.

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

# A  TASK DESCRIPTIONS

The description for each task is provided in Table 3.

| Task Name | Acronym | Description |
|---|---|---|
| Textual Question Answering | TQA | Given a textual context in the source language and a question in the target language, the task involves generating an accurate open-ended answer in the target language based on the provided context. |
| Text Summarization | TSUM | Given a textual context in the source language, the task involves generating a shorter version in the target language that retains the most important information. |
| Machine Translation | MT | Given a textual context, the task involves translating the text from the source language into a different target language, preserving the meaning while adapting to linguistic and grammatical norms. |
| Automatic Speech Recognition | ASR | The task involves converting the spoken language into written text in the same language, focusing on accurate transcription of the speech signal. |
| Spoken Question Answering | SQA | Given a speech context in the source language and a textual question in the target language, the task involves generating an accurate open-ended answer in the target language based on the provided context. |
| Speech Summarization | SSUM | Given a speech context in the source language, the task involves generating a shorter version of the spoken content in the target language that retains the most important information. |
| Speech Translation | ST | The task involves translating a speech in the source language into text in the target language, combining speech recognition and machine translation in a single task. |
| Video Question Answering | VQA | Given a video context in the source language and a textual question in the target language, the task involves generating an accurate open-ended answer in the target language based on the provided context. |
| Video Summarization | VSUM | Given a video context in the source language, the task involves generating a summary in the target language based on the provided context. |
| Audio-Video Recognition | AVR | Given both video and speech contexts, the task involves generating an accurate content transcript. |
| Audio-Video Question Answering | AVQA | Given both video and speech contexts in the source language and a textual question in the target language, the task involves generating an accurate open-ended answer in the target language based on the provided audio-visual context. |
| Audio-Video Summarization | AVSUM | Given both video and speech contexts in the source language, the task involves generating a shorter version of the content in the target language that retains the most important information. |
| Audio-Video Translation | AVT | Given both video and speech contexts, the task involves generating an accurate content translation. |

Table 3: Extended description of the tasks supported by the MCIF benchmark.

# B  DATA CREATION PROCESS AND GUIDELINES

The MCIF gold data comprises English transcripts, QA pairs, and summaries as well as their translations into German, Italian, and Mandarin Chinese. In the following, we present all details about the data creation process.

**Transcripts.**  To produce the English talks' transcripts we relied on a language service agency to hire 2 professional English linguists, who were paid per audio runtime (€3/min, in line

with market rates). Professionals were provided with detailed **guidelines** on how to perform the task (available at: `https://github.com/hlt-mt/mcif/blob/main/dataset_build/annotation_guidelines/Transcription_Guidelines.pdf`). The process began from the ASR outputs (model details are internal to the agency and thus confidential), with professionals revising and correcting the automatic transcripts using MateDub,[4] a CAT-assisted tool that integrates video playback for context-aware transcription. Professionals were instructed to produce clean, fluent transcripts that closely align with the original audio, while respecting technical jargon, US spelling, and proper punctuation. Disfluencies and background noises were omitted. All transcripts were then reviewed by domain specialists who ensured their quality, as well as strict adherence to the task guidelines and correct use of terminology.

**Questions and Answers.** The creation of the original English QA pairs was carried out by MS/PhD students and researchers, who are all experts in NLP and machine learning. All contributors followed a set of detailed **instructions**, fully available at `https://github.com/hlt-mt/mcif/blob/main/dataset_build/annotation_guidelines/Question-Answering_Guidelines.pdf`, which outlined the process and quality criteria for creating and annotating the QA pairs. For each talk, annotators were asked to produce at least 10 QA pairs, divided by both the type of *question* and the type of *answer*. For question types, each talk required: *i)* 3 general questions (pre-assigned, the same for all papers), *ii)* 3 realistic, abstract-based questions, created after reading the abstract, and *iii)* 4 paper-specific, transcript-based questions. We enforced this distribution to ensure a balanced representation of different question types. In all cases, contributors had to annotate each QA pair with a timestamp indicating where the answer appeared in the video, and assign **a label reflecting its source of information**: **A** (answerable from audio only), **V** (from video only), **AV** (from both), or **NA** (not answerable). A target distribution of answer labels was also required for each talk: a minimum of 7 A/AV pairs (with at least 5 AV), 2 V pairs, and 1 NA pair. The guidelines, linked above, provided detailed recommendations on how to formulate clear, specific, and concise questions, avoiding vague or multi-part formulations, and ensuring answers directly addressed the question in no more than two sentences. They also included conventions for US spelling and number formatting.

**Translations.** Transcripts, QA pairs and summaries were all translated by professional translators hired via the same language service agency who provided linguists to create the talks' transcripts. Two professionals were assigned for each target language (German, Italian, and Mandarin Chinese), for a total of 6 professionals. Translations were paid per weighted word count (accounting for repetitions, translation memory matches, and MT suggestions) at an average rate of 0.04€/source word. The process started from an internal MT system (internal to the agency and thus confidential), with translators working in the CAT tool MateCat.[5] They were free to entirely modify or disregard automatic suggestions to ensure adequacy and fluency. Translators adhered to the original English formatting and respected language variants (Italian for Italy, German for Germany, Mandarin for Chinese). Specifically for transcripts' translation, they were instructed *not* to translate *i)* original English paper titles, if any, *ii)* non-English words or expressions that were provided as multilingual examples during the presentation, if any. All produced translations were further revised by domain experts who are native speakers of each target language to ensure data quality, consistency with the guidelines, and terminological correctness.

**Data quality and consistency.x** To summarize, our data creation process ensures data quality by relying on hired professionals, domain experts, and by entailing multiple rounds of conformity revisions. Notably, we also maximize consistency to allow for comparability across several dimensions:

- *Cross-lingual* consistency was ensured by design, as all original content (talks' transcripts, QA pairs, and summaries) was first created in English and subsequently translated into Italian, German, and Mandarine Chinese;
- *Cross-modality* consistency is ensured by definition, as the same gold data are used for a given task across all modalities;
- *Consistency with task guidelines* was maintained through expert verification, during which domain specialists not only assessed the linguistic quality of the professional transcripts

---

[4]`https://matedub.com/`
[5]`https://www.matecat.com/`

and translations but also confirmed strict adherence to task guidelines and the correct use and translation of domain-specific terminology.

## C  LIST OF PROMPTS

The fixed prompts used for MCIF$_{\text{fix}}$ are reported in Table 4. The list of prompts from which we sampled to create MCIF$_{\text{mix}}$ is presented in Tables 5 to 7.

| Lang. | Prompt |
|---|---|
| | *Recognition* |
| 🇺🇸 | Transcribe the English content. |
| | *Translation* |
| 🇩🇪 | Übersetze den englischem Inhalt nach Deutsch. |
| 🇮🇹 | Traduci il contenuto inglese in italiano. |
| 🇨🇳 | 将英文内容翻译成中文。 |
| | *Question Answering* |
| 🇺🇸 | Answer the following question concisely given the English content: {QUESTION} |
| 🇩🇪 | Beantworte die folgende Frage kurz und bündig basierend auf dem englischen Inhalt: {QUESTION} |
| 🇮🇹 | Rispondi in modo conciso alla seguente domanda dato il contenuto inglese: {QUESTION} |
| 🇨🇳 | 根据所给的英文内容，简要回答以下问题：{QUESTION} |
| | *Summarization* |
| 🇺🇸 | Summarize the English content in an abstract of approximately 200 words. |
| 🇩🇪 | Fasse den englischen Inhalt in einem Abstract mit maximal 200 Wörtern zusammen. |
| 🇮🇹 | Riassumi il contenuto inglese in un abstract di circa 200 parole. |
| 🇨🇳 | 用400个字左右概括所给的英语内容。 |

Table 4: Fixed prompts used for MCIF$_{\text{fix}}$.

| Lang. | Prompt |
|---|---|
| | *Recognition* |
| 🇺🇸 | **1.** Transcribe the English content.
**2.** Please write down what is said in the English content.
**3.** Generate a transcription of the English content.
**4.** Convert the English content into text.
**5.** Produce a written version of the English content.
**6.** Provide a text transcript of the English content.
**7.** Accurately transcribe the English content.
**8.** Turn the English content into written text.
**9.** Create a verbatim transcript of the English content.
**10.** Write out the English content as it is stated. |
| | *Translation* |
| 🇩🇪 | **1.** Übersetze den englischem Inhalt nach Deutsch.
**2.** Übersetze den englischen Inhalt ins Deutsche.
**3.** Gib den englischen Inhalt auf Deutsch wieder.
**4.** Übertrage den englischen Inhalt ins Deutsche.
**5.** Führe eine Übersetzung des englischen Inhalts ins Deutsche durch.
**6.** Übersetze den Inhalt aus dem Englischen ins Deutsche.
**7.** Formuliere den englischen Inhalt auf Deutsch.
**8.** Erstelle eine deutsche Übersetzung des englischen Inhalts.
**9.** Übertrage den englischen Inhalt in die deutsche Sprache.
**10.** Gib den englischen Inhalt sinngemäß auf Deutsch wieder. |
| 🇮🇹 | **1.** Traduci il contenuto inglese in italiano.
**2.** Dammi una traduzione in italiano del contenuto in inglese.
**3.** Converti il contenuto inglese in italiano.
**4.** Scrivi una traduzione italiana del contenuto in inglese.
**5.** Traduci in italiano ciò che viene detto in inglese.
**6.** Riporta il contenuto inglese in lingua italiana.
**7.** Fornisci una versione italiana del contenuto inglese.
**8.** Effettua la traduzione del contenuto inglese in italiano.
**9.** Trasforma il contenuto in inglese in una versione italiana.
**10.** Rendi in italiano il contenuto in inglese. |
| 🇨🇳 | **1.** 将英文内容翻译成中文。
**2.** 把英文内容翻译成中文。
**3.** 将所给的英文内容转换成中文。
**4.** 请将所给出的英文翻译成中文。
**5.** 将该段英文内容翻译为中文。
**6.** 将这段英语内容表达为中文。
**7.** 用中文翻译所给内容中的英文。
**8.** 请将英文内容转换为汉语。
**9.** 收到英文内容后，用中文表述其意思。
**10.** 将这段英语内容用中文重新表达。 |

Table 5: List of prompts sampled to create MCIF$_{\text{mix}}$ (part 1).

| Lang. | Prompt |
|---|---|
| | *Question Answering* |
| 🇺🇸 | **1.** Answer the following question concisely given the English content: {QUESTION}
**2.** Based on the English content, respond to this question with a brief answer: {QUESTION}
**3.** Use the English content to provide a concise answer to the question below: {QUESTION}
**4.** Consider the English content and provide a brief reply to the question: {QUESTION}
**5.** Given the English content, what is a concise answer to the question: {QUESTION}
**6.** Relying on the English content, provide a concise answer: {QUESTION}
**7.** Interpret the English content and concisely respond to the following: {QUESTION}
**8.** Consider the English content and briefly answer this: {QUESTION}
**9.** Use the content in English to formulate a concise response: {QUESTION}
**10.** Refer to the English content to answer the question. Be concise: {QUESTION} |
| 🇩🇪 | **1.** Beantworte die folgende Frage kurz und bündig basierend auf dem englischen Inhalt: {QUESTION}
**2.** Beantworte folgende Frage kurz und bündig unter Bezugnahme auf den englischen Inhalt: {QUESTION}
**3.** Verwende den englischen Inhalt, um diese Frage kurz und bündig zu beantworten: {QUESTION}
**4.** Beziehe dich auf den englischen Inhalt an und gib eine kurze Antwort auf die Frage: {QUESTION}
**5.** Basierend auf dem englischen Inhalt, beantworte die nachfolgende Frage kurz und bündig: {QUESTION}
**6.** Nutze den englischen Inhalt zur knappen Beantwortung der Frage: {QUESTION}
**7.** Analysiere den englischen Inhalt und beantworte die Frage kurz und bündig: {QUESTION}
**8.** Beantworte diese Frage kurz und bündig mithilfe des englischen Inhalts: {QUESTION}
**9.** Analysiere den englischen Inhalt und beantworte dann diese Frage kurz und bündig: {QUESTION} Orientiere dich am englischen Inhalt und gib eine kurze Antwort:
**10.** {QUESTION} |
| 🇮🇹 | **1.** Rispondi in modo conciso alla seguente domanda dato il contenuto inglese: {QUESTION}
**2.** Rispondi brevemente alla seguente domanda utilizzando il contenuto inglese: {QUESTION}
**3.** Esamina il contenuto inglese e rispondi alla domanda in modo conciso: {QUESTION}
**4.** Fornisci una breve risposta alla domanda basandoti sul contenuto inglese: {QUESTION}
**5.** Considera il contenuto inglese e rispondi sinteticamente a questa domanda: {QUESTION}
**6.** Rispondi alla domanda in modo conciso servendoti del contenuto inglese: {QUESTION}
**7.** Sulla base del contenuto inglese, dai una risposta concisa alla domanda: {QUESTION}
**8.** Rispondi sinteticamente alla domanda usando le informazioni del contenuto inglese: {QUESTION}
**9.** Considera il contenuto inglese per rispondere alla seguente domanda in maniera concisa: {QUESTION}
**10.** Utilizza il contenuto inglese come base per rispondere. Fornisci una risposta concisa: {QUESTION} |
| 🇨🇳 | **1.** 根据所给的英文内容，简要回答以下问题：{QUESTION}
**2.** 根据英语内容，简要回答下面的问题：{QUESTION}
**3.** 接收到英文内容后，简短回答以下问题：{QUESTION}
**4.** 请结合英语内容，对如下问题简要作答：{QUESTION}
**5.** 根据所给英文内容，给出简短的答案：{QUESTION}
**6.** 请基于所给内容中的英文信息简要回答问题：{QUESTION}
**7.** 听完英语内容后，请为以下提问简要作答：{QUESTION}
**8.** 参考英语内容，简要回答下列问题：{QUESTION}
**9.** 使用所给内容中的英文来简要回答问题：{QUESTION}
**10.** 请依据英文内容简要回答问题：{QUESTION} |

Table 6: List of prompts sampled to create MCIF$_{mix}$ (part 2).

| Lang. | Prompt |
|---|---|
| | *Summarization* |
| 🇺🇸 | **1.** Summarize the English content in an abstract of approximately 200 words. **2.** Provide a summary of the English content using roughly 200 words. **3.** Condense the English content into a summary of about 200 words. **4.** Write a brief summary (about 200 words) of the English content. **5.** Summarize the English content, keeping it around 200 words. **6.** Create a concise summary of the English content in about 200 words. **7.** Using approximately 200 words, summarize the English audio content. **8.** Capture the main points of the English content in about 200 words. **9.** Give a summary of approximately 200 words of the English content. **10.** Write a short summary (about 200 words) of what's in the English content. |
| 🇩🇪 | **1.** Fasse den englischen Inhalt in einem Abstract mit ungefähr 200 Wörtern zusammen. **2.** Fasse den englischen Inhalt in ungefähr 200 Wörtern zusammen. **3.** Erstelle eine Zusammenfassung (um die 200 Wörter) des englischen Inhalts. **4.** Schreibe eine kurze Zusammenfassung des englischen Inhalts mit ungefähr 200 Wörtern. **5.** Gib den englischen Inhalt in ca. 200 Wörtern wieder. **6.** Fasse den Inhalt auf Englisch in ungefähr 200 Wörtern zusammen. **7.** Verfasse eine ungefähr 200 Wörter lange Zusammenfassung des englischen Inhalts. **8.** Erstelle eine kompakte Zusammenfassung des englischen Inhalts in ungefähr 200 Wörtern. **9.** Gib eine Kurzfassung des englischen Inhalts in ca. 200 Wörtern. **10.** Formuliere eine Zusammenfassung des englischen Inhalts mit ungefähr 200 Wörtern. |
| 🇮🇹 | **1.** Riassumi il contenuto inglese in un abstract di circa 200 parole. **2.** Riassumi il contenuto inglese in circa 200 parole. **3.** Fai un riassunto del contenuto in inglese con circa 200 parole. **4.** Scrivi un breve riassunto del contenuto inglese (circa 200 parole). **5.** Sintetizza il contenuto inglese in circa 200 parole. **6.** Riassumi quanto detto nel contenuto inglese usando circa 200 parole. **7.** Rendi in sintesi il contenuto inglese (circa 200 parole). **8.** Scrivi un riassunto in circa 200 parole dell'audio inglese. **9.** Esprimi in forma sintetica il contenuto inglese (circa 200 parole). **10.** Fornisci una sintesi del contenuto audio inglese in circa 200 parole. |
| 🇨🇳 | **1.** 用400个字左右概括所给的英语内容。 **2.** 将英文内容用400个字概括。 **3.** 请用400字左右总结这段英文内容的要点。 **4.** 对这段英文内容做出400字左右的简要概括。 **5.** 用大约400个汉字总结这段英文内容。 **6.** 将这段英语内容的核心内容简要描述（400字左右）。 **7.** 以简洁语言，约400字总结英文内容。 **8.** 提炼英文内容的主要信息，用400字左右表达。 **9.** 用大约400字写出这段英文内容的总结。 **10.** 对所给的英语内容进行400字左右的总结。 |

Table 7: List of prompts sampled to create MCIF$_{\text{mix}}$ (part 3).

## D   MODELS

The models used for the analyses are listed in Table 8 (next page) and run using the HuggingFace Transformer version indicated for each model, as some of them require a specific version.

For all models, we use the default generation parameters, following the usage instructions reported in the model cards. When available, we adopt the suggested system prompts for each model, with the additional instruction: "Only return the answer requested. Do not include any explanation or introductions." The maximum number of new tokens is set to 4096 for all models. The code used for inference is released upon paper acceptance. The inference is performed using a single NVIDIA GH200 120GB GPU.

| Model | Param. | In.Mod. | Weights | HFv |
|---|---|---|---|---|
| Aya Expanse (Dang et al., 2024) | 8B | 📄 | `https://huggingface.co/CohereLabs/aya-expanse-8b` | 4.51.3 |
| Gemma 3 (Team Gemma et al., 2025) | 12B | 📄 | `https://hf.co/google/gemma-3-12b-it` | 4.51.3 |
| Llama 3.1 (Grattafiori et al., 2024) | 8B | 📄 | `https://hf.co/meta-llama/Llama-3.1-8B-Instruct` | 4.51.3 |
| GPT-oss (OpenAI et al., 2025) | 20B | 📄 | `https://huggingface.co/openai/gpt-oss-20b` | 4.55.0 |
| Phi4 (Abdin et al., 2024a) | 14.7B | 📄 | `https://hf.co/microsoft/phi-4` | 4.51.3 |
| Qwen3 (Yang et al., 2025a) | 14B | 📄 | `https://huggingface.co/Qwen/Qwen3-14B` | 4.51.3 |
| Tower-Plus (Rei et al., 2025) | 9B | 📄 | `https://huggingface.co/Unbabel/Tower-Plus-9B` | 4.51.3 |
| DeSTA2 (Lu et al., 2024b) | 8B | 🗣️ | `https://hf.co/DeSTA-ntu/DeSTA2-8B-beta` | 4.51.3 |
| GraniteSpeech 3.3 (Saon et al., 2025) | 8B | 🗣️ | `https://hf.co/ibm-granite/granite-speech-3.3-8b` | 4.52.4 |
| Phi4-Multimodal (Abdin et al., 2024b) | 5.6B | 🗣️ | `https://hf.co/microsoft/Phi-4-multimodal-instruct` | 4.48.2 |
| Qwen2-Audio (Chu et al., 2024) | 7B | 🗣️ | `https://hf.co/Qwen/Qwen2-Audio-7B-Instruct` | 4.51.3 |
| UltraVox 0.5[†] | 8.07B | 🗣️ | `https://hf.co/fixie-ai/ultravox-v0_5-llama-3_2-1b` | 4.51.3 |
| InternVL3 (Chen et al., 2024b) | 14B | 🎞️ | `https://huggingface.co/OpenGVLab/InternVL3-14B` | 4.51.3 |
| LLaVA-NeXT-Video (Zhang et al., 2024b) | 7B | 🎞️ | `https://huggingface.co/llava-hf/LLaVA-NeXT-Video-7B-hf` | 4.51.3 |
| Qwen2.5-VL (Wang et al., 2024a) | 7B | 🎞️ | `https://huggingface.co/Qwen/Qwen2.5-VL-7B-Instruct` | 4.51.3 |
| VideoLLaMA3 (Zhang et al., 2025) | 7B | 🎞️ | `https://huggingface.co/DAMO-NLP-SG/VideoLLaMA3-7B` | 4.51.3 |
| Video-XL-2 (Shu et al., 2024) | 8B | 🎞️ | `https://huggingface.co/BAAI/Video-XL-2` | 4.51.3 |
| Gemma 3n [‡] | 4B | 📄🗣️🎞️ | `https://huggingface.co/google/gemma-3n-E4B-it` | 4.53.0 |
| Ming-Lite-Omni (AI, 2025) | 2.8B | 📄🗣️🎞️ | `https://huggingface.co/inclusionAI/Ming-Lite-Omni` | 4.45.0 |
| MiniCPM-o-2 (Yao et al., 2024) | 8B | 📄🗣️🎞️ | `https://huggingface.co/openbmb/MiniCPM-o-2_6` | 4.44.2 |
| Ola (Liu et al., 2025) | 7B | 📄🗣️🎞️ | `https://huggingface.co/THUdyh/Ola-7b` | 4.43.4 |
| Qwen2.5-Omni (Xu et al., 2025a) | 7B | 📄🗣️🎞️ | `https://hf.co/microsoft/Phi-4-multimodal-instruct` | 4.51.3 |

Table 8: Details of the models, including the number of parameters (**Param.**), input modalities analyzed in this paper (**In.Mod.**), their public weights release (**Weights**), and the HuggingFace Transformer version (**HFv**) used for the experiments on the MCIF benchmark. [†] `https://www.ultravox.ai/` [‡] `https://deepmind.google/models/gemma/gemma-3n/`

# E  EXTENDED RESULTS

The scores of MCIF$_{\text{fix}}$ and MCIF$_{\text{mix}}$ per language are presented in Tables 9 and 10.

| Context | Model | Input Modality | REC. WER↓ 🇺🇸 | TRANSLATION COMET↑ 🇩🇪 | 🇮🇹 | 🇨🇳 | QUESTION ANSW. BERTSCORE↑ 🇺🇸 | 🇩🇪 | 🇮🇹 | 🇨🇳 | SUMMARIZATION BERTSCORE↑ 🇺🇸 | 🇩🇪 | 🇮🇹 | 🇨🇳 |
|---|---|---|---|---|---|---|---|---|---|---|---|---|---|---|
| SHORT | DeSTA2 | 🗣 | 54.0 | 72.5 | 76.7 | 76.7 | 23.7 | 24.7 | 17.9 | 2.4 | | | | |
| | GraniteSpeech | | 9.4 | 42.0 | 52.2 | 62.1 | 11.8 | 1.5 | 0.4 | -11.6 | | | | |
| | Phi4-Multimodal | | **6.8** | **77.7** | **81.2** | **81.6** | 42.3 | 33.0 | 32.9 | 40.0 | ✕ | | | |
| | Qwen2-Audio | | 31.7 | 71.6 | 73.9 | 79.3 | 33.3 | 30.5 | 30.6 | 36.1 | | | | |
| | UltraVox v0.5 | | 127.7 | 45.9 | 50.6 | 33.5 | 16.1 | 22.8 | 16.8 | 22.9 | | | | |
| | InternVL3 | 🎞 | ✕ | | ✕ | | 30.9 | 28.6 | 30.0 | 37.4 | | | | |
| | LLaVA-NeXT | | | | | | 15.3 | 10.3 | 8.8 | 20.3 | | | | |
| | Qwen2.5-VL | | | | | | 34.3 | 38.9 | 38.8 | **44.5** | ✕ | | | |
| | VideoLLaMA3 | | | | | | 16.7 | 31.2 | 20.0 | 28.7 | | | | |
| | Video-XL2 | | | | | | 16.7 | 14.2 | 12.1 | 11.4 | | | | |
| | Gemma 3n | 🗣🎞 | 35.1 | 70.6 | 74.2 | 74.3 | 25.5 | 27.5 | 26.6 | 25.0 | | | | |
| | Ming-Lite-Omni | | 117.5 | 55.8 | 55.9 | 47.4 | 21.0 | 14.6 | 14.6 | 13.1 | | | | |
| | MiniCPM-o-2 | | 144.8 | 35.0 | 41.3 | 42.9 | 23.0 | 18.8 | 18.6 | 25.0 | ✕ | | | |
| | Ola | | 104.1 | 72.5 | 76.9 | 80.4 | 33.3 | 37.2 | **39.3** | 39.5 | | | | |
| | Qwen2.5-Omni | | 43.5 | 74.2 | 76.4 | 81.2 | 35.8 | 35.5 | 34.0 | 32.0 | | | | |
| | *Gemini 2.5 Flash* | | 14.9 | 62.7 | 65.7 | 72.6 | **45.5** | **41.6** | 37.6 | 37.7 | | | | |
| LONG | Aya Expanse | 📄 | ✕ | 62.5 | 68.6 | 74.9 | 28.1 | 28.9 | 25.2 | 24.4 | 16.7 | 22.7 | **24.6** | **36.4** |
| | Gemma 3 | | | 82.3 | **87.9** | **86.3** | 29.1 | 26.7 | 25.2 | 10.4 | 21.6 | 11.6 | 12.6 | -7.7 |
| | GPT-oss | | | 72.0 | 78.8 | 74.4 | 21.0 | 24.6 | 22.2 | 30.6 | 11.0 | 15.8 | 12.0 | 32.4 |
| | Llama 3.1 | | | 80.5 | 84.3 | 79.3 | 29.7 | 31.3 | 29.4 | 30.7 | **22.5** | 16.7 | 23.9 | 36.1 |
| | Phi4 | | | 83.0 | 85.7 | 84.8 | 32.5 | 32.5 | 33.0 | 25.1 | 20.1 | **23.8** | 16.0 | -7.7 |
| | Qwen3 | | | 82.5 | 86.4 | 85.4 | 37.9 | 40.7 | 36.3 | 36.8 | 22.4 | 12.5 | 22.6 | 22.3 |
| | Tower+ | | | **83.6** | 87.3 | 85.9 | 30.2 | 31.8 | 29.7 | 26.4 | 18.9 | 11.4 | 19.6 | 29.2 |
| | DeSTA2 | 🗣 | 112.9 | 39.9 | 43.7 | 40.4 | 18.3 | 18.3 | 16.0 | -2.5 | 7.5 | 6.4 | 7.5 | -7.7 |
| | GraniteSpeech | | 99.9 | 35.4 | 40.3 | 32.3 | -22.3 | -26.1 | -25.7 | -20.5 | -7.0 | -10.0 | -10.0 | -14.7 |
| | Phi4-Multimodal | | 39.2 | 56.3 | 66.4 | 56.5 | 39.1 | 36.0 | 33.8 | 41.6 | 18.0 | 19.7 | 18.4 | 14.8 |
| | Qwen2-Audio | | 92.9 | 39.3 | 43.2 | 40.5 | 28.9 | 27.7 | 26.9 | 31.5 | 3.0 | 5.8 | 10.1 | 9.2 |
| | UltraVox v0.5 | | 89.1 | 36.8 | 40.8 | 36.4 | 21.4 | 12.1 | 4.2 | 13.2 | 6.4 | -4.9 | -3.7 | -9.7 |
| | InternVL3 | 🎞 | ✕ | | ✕ | | 26.0 | 27.0 | 26.1 | 31.4 | 15.6 | 14.8 | 19.1 | 32.2 |
| | LLaVA-NeXT | | | | | | 9.7 | -1.5 | 1.7 | 18.7 | -6.9 | -5.3 | -6.3 | -9.5 |
| | Qwen2.5-VL | | | | | | 25.7 | 36.3 | 36.0 | 37.0 | 19.4 | 13.6 | 24.2 | 35.9 |
| | VideoLLaMA3 | | | | | | 22.9 | 31.4 | 20.7 | 32.2 | -16.5 | -17.8 | -21.7 | -23.7 |
| | Video-XL2 | | | | | | 20.1 | 15.7 | 14.9 | 18.1 | 9.6 | 6.5 | 4.1 | -5.4 |
| | MiniCPM-o-2 | 🗣🎞 | 170.6 | 24.6 | 24.9 | 25.2 | -48.2 | -35.4 | -38.2 | -30.3 | -59.7 | -39.1 | -37.1 | -20.6 |
| | Ola | | 14.0 | 60.3 | 73.4 | 55.8 | 32.4 | 38.0 | 38.7 | 35.6 | 18.2 | 11.2 | 12.4 | 7.3 |
| | Qwen2.5-Omni | | 98.5 | 37.2 | 42.3 | 40.8 | 34.4 | 26.3 | 26.3 | 43.0 | 17.3 | 10.4 | 11.9 | -3.9 |
| | *Gemini 2.5 Flash* | | **11.9** | 78.1 | 79.1 | 71.9 | **47.8** | **43.7** | **46.3** | **46.6** | 16.7 | 20.8 | 23.1 | 35.6 |

Table 9: Results on MCIF$_{\text{fix}}$ of the 23 models for each input modality, context, task (divided into the four macro-tasks: RECOGNITION, TRANSLATION, QUESTION-ANSWERING, and SUMMA-RIZATION), and target language (🇺🇸 for English, 🇩🇪 for German, 🇮🇹 for Italian, and 🇨🇳 for Chinese, while source language, being always English, is omitted). The best result by context is marked in **bold**, and the best overall result is underlined. Gemma 3n and Ming-Lite-Omni are removed from LONG scores as they are not able to process long-form speech.

| Context | Model | Input Modality | REC. WER↓ 🇺🇸 | TRANSLATION COMET↑ 🇩🇪 | 🇮🇹 | 🇨🇳 | QUESTION ANSW. BERTSCORE↑ 🇺🇸 | 🇩🇪 | 🇮🇹 | 🇨🇳 | SUMMARIZATION BERTSCORE↑ 🇺🇸 | 🇩🇪 | 🇮🇹 | 🇨🇳 |
|---|---|---|---|---|---|---|---|---|---|---|---|---|---|---|
| SHORT | DeSTA2 | 🗣 | 83.0 | 72.4 | 76.5 | 76.6 | 26.3 | 25.1 | 22.1 | 0.8 | × | | | |
| | GraniteSpeech | | 9.5 | 42.9 | 49.4 | 47.5 | 11.4 | 0.7 | 0.7 | -11.4 | | | | |
| | Phi4-Multimodal | | **6.7** | **77.7** | **81.3** | **81.3** | **43.4** | 37.0 | 35.5 | 33.7 | | | | |
| | Qwen2-Audio | | 31.9 | 71.3 | 73.8 | 78.8 | 34.1 | 32.3 | 31.0 | 33.6 | | | | |
| | UltraVox v0.5 | | 172.6 | 43.4 | 43.2 | 43.1 | 18.9 | 21.0 | 17.9 | 18.4 | | | | |
| | InternVL3 | 🎞 | × | × | | | 28.8 | 28.7 | 28.8 | 38.9 | × | | | |
| | LLaVA-NeXT | | | | | | 12.2 | 8.8 | 8.2 | 19.4 | | | | |
| | Qwen2.5-VL | | | | | | 33.9 | 39.2 | 37.6 | **40.5** | | | | |
| | VideoLLaMA3 | | | | | | 22.3 | 25.8 | 21.6 | 25.5 | | | | |
| | Video-XL2 | | | | | | 17.6 | 15.6 | 13.4 | 7.8 | | | | |
| | Gemma 3n | 🗣🎞 | 58.9 | 70.7 | 71.3 | 72.3 | 30.4 | 26.0 | 26.1 | 17.7 | × | | | |
| | Ming-Lite-Omni | | 128.2 | 57.3 | 55.1 | 47.5 | 15.4 | 16.2 | 10.7 | 11.0 | | | | |
| | MiniCPM-o-2 | | 207.1 | 34.5 | 40.0 | 41.8 | 19.6 | 24.7 | 20.4 | 27.7 | | | | |
| | Ola | | 98.8 | 72.5 | 76.4 | 80.1 | 34.3 | 36.1 | 37.8 | 39.9 | | | | |
| | Qwen2.5-Omni | | 48.0 | 74.0 | 74.4 | 81.0 | 36.9 | 35.9 | 35.4 | 32.0 | | | | |
| | *Gemini 2.5 Flash* | | 12.8 | 67.4 | 68.2 | 72.0 | 40.9 | **39.3** | **39.1** | 38.6 | | | | |
| LONG | Aya Expanse | 📄 | × | 63.3 | 70.1 | 72.9 | 20.9 | 24.1 | 23.0 | 24.5 | 15.2 | **21.8** | **24.0** | **35.5** |
| | Gemma 3 | | | 81.6 | 86.0 | 82.7 | 30.6 | 25.3 | 25.4 | 6.1 | 19.1 | 11.4 | 12.6 | -5.3 |
| | GPT-oss | | | 66.7 | 73.5 | 70.2 | 18.3 | 22.6 | 21.8 | 21.0 | 7.7 | 13.7 | 12.9 | 29.3 |
| | Llama 3.1 | | | 79.5 | 82.1 | 76.8 | 31.0 | 30.6 | 30.2 | 32.2 | 19.9 | 19.6 | 23.3 | 35.3 |
| | Phi4 | | | 82.0 | **86.9** | 85.1 | 31.8 | 31.7 | 32.5 | 22.5 | 18.7 | 21.4 | 17.4 | 0.6 |
| | Qwen3 | | | **82.8** | 85.8 | 84.9 | 35.4 | 39.2 | 35.1 | 32.6 | **20.8** | 19.4 | 23.7 | 16.3 |
| | Tower+ | | | 81.4 | 83.5 | **86.0** | 24.9 | 26.8 | 29.8 | 12.2 | 18.2 | 15.0 | 21.9 | 15.5 |
| | DeSTA2 | 🗣 | 132.5 | 39.5 | 43.2 | 39.7 | 17.8 | 18.3 | 16.8 | -2.6 | 4.7 | 5.6 | 7.9 | -6.8 |
| | GraniteSpeech | | 80.4 | 35.1 | 39.4 | 29.4 | -21.9 | -25.3 | -24.2 | -19.9 | -6.8 | -11.0 | -9.4 | -14.6 |
| | Phi4-Multimodal | | 29.8 | 60.7 | 65.4 | 52.4 | 39.5 | 35.8 | 34.4 | 39.6 | 17.6 | 18.0 | 18.1 | 17.8 |
| | Qwen2-Audio | | 93.1 | 39.5 | 43.4 | 40.5 | 29.5 | 28.5 | 28.3 | 29.5 | 2.1 | 6.4 | 10.0 | 6.5 |
| | UltraVox v0.5 | | 92.5 | 36.7 | 40.6 | 36.7 | 21.9 | 8.2 | 8.9 | 11.1 | 6.1 | -4.6 | -5.2 | -8.7 |
| | InternVL3 | 🎞 | × | × | | | 25.9 | 26.9 | 24.6 | 34.1 | 13.7 | 17.9 | 18.1 | 31.9 |
| | LLaVA-NeXT | | | | | | 3.2 | -1.8 | 3.4 | 16.2 | -9.6 | -7.2 | -6.2 | -3.7 |
| | Qwen2.5-VL | | | | | | 31.9 | 36.8 | 37.6 | 33.3 | 17.2 | 18.7 | 21.0 | 23.8 |
| | VideoLLaMA3 | | | | | | 24.1 | 30.7 | 24.0 | 27.2 | -3.7 | -31.4 | -25.2 | -71.9 |
| | Video-XL2 | | | | | | 21.7 | 18.4 | 16.5 | 13.0 | 7.5 | 1.4 | 0.1 | 7.6 |
| | MiniCPM-o-2 | 🗣🎞 | 179.2 | 25.0 | 26.6 | 26.0 | -47.5 | -38.2 | -37.2 | -30.7 | -59.0 | -39.5 | -39.4 | -20.8 |
| | Ola | | 36.8 | 47.1 | 59.1 | 48.5 | 36.1 | 33.5 | 34.8 | 32.4 | 15.3 | 10.9 | 10.9 | 18.1 |
| | Qwen2.5-Omni | | 94.9 | 37.5 | 41.6 | 41.6 | 41.3 | 31.4 | 31.0 | 35.7 | 8.8 | 10.8 | 10.5 | 7.5 |
| | *Gemini 2.5 Flash* | | 7.9 | 75.2 | 81.2 | 83.3 | **45.3** | **44.9** | **46.5** | **47.0** | 16.0 | 19.7 | 20.7 | 30.6 |

Table 10: Results on MCIF_mix of the 23 models for each input modality, context, task (divided into the four macro-tasks: RECOGNITION, TRANSLATION, QUESTION ANSWERING, and SUMMARIZATION), and target language (🇺🇸 for English, 🇩🇪 for German, 🇮🇹 for Italian, and 🇨🇳 for Chinese, while source language, being always English, is omitted). The best result by context is marked in **bold**, and the best overall result is underlined. Gemma 3n and Ming-Lite-Omni are removed from LONG scores as they are not able to process long-form speech.

## F  FAILURE EXAMPLES

Examples of the failure cases of the benchmarked systems mentioned in Section 5 are presented in Table 11.

Table 11: Examples of models failures on the MCIF benchmark (non-exhaustive).

| # | Model | Task | Output |
|---|-------|------|--------|
| 1 | Ola, LLaVa-NeXT | **REC** **SUM** | **Transcription output contains only a table entry**:
`Main findings 85 FTw COSINE L2R MLC 80 75 70 65 60 55 50 All weak labels 40 30 20 10 0 5 10 20 30 40 50 All Validation Performance Delta (%) FTw COSINE L2R MLC Adapterc FTc LoRAc BiFiTc Adapterc`
**Output contains the description of a single slide instead of summarizing the video**:
`The image displays a graphical representation of a model's performance on a dataset, showing the model's accuracy across different layers and the overall accuracy.  It highlights that the model's performance improves as it goes deeper into the layers, with the highest accuracy at the base layer and decreasing as it moves towards the top layer.  The model's performance is also compared to a baseline model, which has a lower accuracy.` |
| 2 | Qwen2-Audio | **REC** **TRANS** | **The transcription of a whole talk contains only the first part:**
`The original content of this audio is:'Hi, my name is Matthias Lendermann, and today I am going to give you a brief introduction to our paper on compositional generalisation without trees using multisets tagging and latent permutations.  This is joint work with my advisors Alexander Colla and Ivan Tovstilov.  Compositional generalisation can be understood as the ability of a learner to handle deeper recursion and unseen compositions.'`
**The translation of a whole talk contains only the first part:**
`Hallo und willkommen zu unserer Präsentation von DeepL, einem neuen Modul für die deutsche Texterkennung auf Dokumentebene und auf Satzebene.  Mein Name ist Regina Stodden und ich werde Ihnen in der ersten Teil des Vortrags führen.  Lassen Sie uns zunächst definieren, was Textsimplifizierung ist.  Textsimplifizierung ist ein Prozess, bei dem ein Text angepasst wird, um seine Verständlichkeit für einen bestimmten Zielgruppe zu verbessern.  Wie Menschen lesen.` |

Table 11: Examples of models failures on the MCIF benchmark (non-exhaustive). (Continued)

| 3 | Gemini 2.5 Flash | **TRANS** | **Output contains an over-elaboration on audio-visual content:**
`SPECIAL INSTRUCTION: I will first think step by step how to translate the audio and then apply the thinking to the translation. Internal monologue:  The user wants me to translate the English audio into German.  I need to listen carefully to the speaker's words and translate them accurately while maintaining the original meaning and` [...]
`"Jedoch kann dieser Effekt gemildert werden, äh, wenn, ähm, äh, wenn` [...]
**Output contains an hallucination on audio-visual content:**
`1.  Semplificazione del Testo Cosa, perché e come?`
`**Testo parlato:** Ciao` [...] |
|---|---|---|---|
| 4 | GraniteSpeech, LLaVA-NeXT, Tower+, Phi4-Multimodal | **TRANS** **QA** **SUM** | **Output in English while prompted for translation in Chinese:**
`hi welcome to our presentation of deplain a new corpus for german text semplification` [...]
**Output in English while prompted for QA in Italian:**
`The Transformer model is known for its efficiency and effectiveness in handling sequential data, which` [...]
**Output in English while prompted for summarization in German**:
`We present CREST (A Joint Framework for Rationalization and Counterfactual Text Generation), a novel approach that` [...]
**Output in English while prompted for summarization in Chinese**:
`Shanshan Chen, a master student from Shenzhen University, presented her work on efficient open domain question answering at ACL 2023. The work focuses on a two-stage model for question answering,` [...] |
| 5 | GraniteSpeech | **QA** | **Answer to the question "Which model did they use to obtain the 82%-87% accuracy?" is a random text:**
`the ldt corpus has 6000 alternative questions across three domains and it has 42000 indirect referring expressions results with t5x-large model are summarized below`
**Full transcript as response to "What are the main data sources for language models?"**
`hi i'm jianbin phd student at university of washington today i'm presenting` [...] `i think that's pretty much all i have for today thank you for your time` |
| 6 | UltraVox v0.5 | **QA** | **False detection of harmful request or denial of responding. E.g., for "用400个字左右概括所给的英语内容。" in different samples the responses are:**
*1)*    `I cannot provide information on illegal activities.  Is there anything else I can help you with?`
*2)* `I cannot fulfill your request.` |

