# OpenReview forum: "MCIF: Multimodal Crosslingual Instruction-Following Benchmark from Scientific Talks"
_ICLR.cc/2026/Conference — ICLR 2026 Poster_

### Official Review · Reviewer_2tAT · 2025-10-26

**Soundness:** 4
**Presentation:** 4
**Contribution:** 3
**Rating:** 8
**Confidence:** 3

**Summary:**

This paper proposes a new benchmark for evaluation of instruction-following capabilities under a comprehensive context of multi-modal, cross-liingual, and varied input length. The benchmark is validated on 23 models, which is good for comparison, and provides guidance on future development.

**Strengths:**

* This benchmark is indeed covering a wide arange of topics, tasks, and languages. It could be used as a supplement to the existing benchmarks that challenge the IF capabilities of MLLMs.
* The paper is generally well-written, easy to follow with pretty illustrations.
* The detailed descriptions on manual annotations are provided for guaranteeing the quality of the benchmark.

**Weaknesses:**

* The definition of "scientific talks" might be limited in the ACL context, which might not be that broad enough to include nature science/medicine domains.
* The evaluation metrics might not be comprehensive enough where recent LLM-as-a-Judge-based metrics are under-explored.
* The prompt templates (see Table 4,5,6) might not be varied enough to be practical. The dynamic prompting by user-simulation (LLM-driven) is not considered and there might exists a risk of over-fitting where future MLLMs might overfit the patterns inherent in such fixed, less-varied templates.
* Discussions and analyses are not enlightening enough.

**Questions:**

* Perhaps modify the descriptions about the scientific talks to be clear, explicit in the introduction.
* The authors are encouraged to extend more metrics as a complement to existing rule-based computing metrics.
* The user-simulation by LLMs is important to an evergreen benchmark which prevents from hacking. The authors are encouraged to provide such prompt augmentation.
* The discussions should contain explicit take-home messages which help the development of future MLLMs. In addition, the case-by-case (especially typical cases in each task) analysis should be provided to be intuitive.
* The authors might consider citation of recent instruction-following studies from the perspective of data recipe:
e.g., Qin, Y., Yang, Y., Guo, P., Li, G., Shao, H., Shi, Y., ... & Sun, X. (2024). Unleashing the power of data tsunami: A comprehensive survey on data assessment and selection for instruction tuning of language models. arXiv preprint arXiv:2408.02085.

---

> ### Author Response · Authors · 2025-11-19
>
> We thank the Reviewer for their positive assessment and for recognizing the value of our human-curated benchmark. We address the raised concerns below:
>
> # Weaknesses
> ## W1. Definition of "scientific talks"
> We thank the Reviewer for the suggestion, and we have clarified that the scientific talks mostly relate to NLP/AI topics in our updated version of the paper. Specifically, in lines 23 and 158 of the revised PDF.
> ## W2. Evaluation metrics might not be comprehensive
> We acknowledge that LLM-as-a-Judge-based metrics are becoming increasingly popular across different tasks. Regarding our cases: *i)* WER is a standard metric for recognition, and we do not see motivations for employing LLM-based metrics, as it is commonly employed in popular evaluation frameworks (e.g., [Dynamic-SUPERB](https://openreview.net/forum?id=s7lzZpAW7T)); *ii)* COMET is the most widespread neural metric for evaluating translation and [the recent dedicated evaluation campaign at WMT](https://www2.statmt.org/wmt25/pdf/2025.wmt-1.23.pdf) has shown that for sentence-based evaluation (which is our case), LLM-as-a-judge metrics underperform traditional metrics, while they excel when it comes to document-wise evaluation; iii) For QA and summarization, we agree that LLM-as-a-judge is an interesting solution, but [recent work on summarization](https://aclanthology.org/2025.iwslt-1.2/) has shown that BERTScore better correlates with human judgements than LLM-as-a-judge and the reproducibility of scores with LLM-as-a-judge is challenging, especially if employing powerful closed-source models that may not be available on the long term. Moreover, there is no standard approach for evaluating these tasks with LLM-judge approaches, especially in audiovisual settings. For these reasons, to ensure comparability with the baselines for future works on our benchmark, we opted for metrics (which, apart from WER, are not rule-based) whose reproducibility in the long term is ensured. Indeed, we agree that improving evaluation metrics is an important topic for future work.
> ## W3. Fixed prompt templates
> We agree with the Reviewer that the variety of the prompt templates may be increased. However, all our prompts are human-curated, so they should reflect how a human would prompt the LLM for the task. While we agree that in the long term there might be the risk of overfitting to the provided prompts, this may be true with any kind of prompt. We see as only possible solution to keep updating the benchmark with newer versions featuring updated content and prompts.
> ## W4. Discussions and analyses are not enlightening
> As mentioned in our Conclusions, our analyses highlight that the major limitations of current systems regard the *“joint speech and video modality integration, long-form processing, and summarization”*. Accordingly, we believe that, as take-home messages to help the development of future MLLMs, future systems development should: *i)* design better strategies to enable processing multiple modalities at once rather than a single modality at a time and, and *ii)* develop systems robust to long-form content by means of either effective modality-specific sequence compression methods or increasing the ability to effectively attend a large context window in the LLM. We have made these recommendations explicit in the Conclusions (Section 6) of the revised PDF.
>
> Finally, we thank the Reviewer for their insightful observations, including the suggested reference, which has been incorporated to improve our work.

---

> > ### Comment · Reviewer_2tAT · 2025-11-26
> >
> > Thank you for the explanations.

---

### Official Review · Reviewer_sQwM · 2025-10-29

**Soundness:** 3
**Presentation:** 3
**Contribution:** 3
**Rating:** 6
**Confidence:** 3

**Summary:**

The paper presents MCIF, a new benchmark to test how well large multimodal models can follow instructions across languages and input types. The dataset comes from real scientific talks with aligned text, audio, and video in four languages: English, German, Italian, and Chinese. It covers 13 tasks grouped into recognition, translation, question answering, and summarization. The authors also test 23 different models including text-only, speech, video, and multimodal ones. The results show that models still struggle a lot when dealing with long inputs, with combining speech and video, and when asked to summarize complex content.

**Strengths:**

1) The idea is very timely. Multimodal and multilingual instruction following is clearly the next big step for LLMs.

2) The dataset design is thoughtful. Everything is aligned across languages and modalities, which makes comparisons fair and controlled.

3) It is fully human-annotated. The effort to manually create transcripts, translations, and QA pairs really increases reliability.

4) The analysis is broad and systematic. The paper looks at different model types, context lengths, and prompt variations. The findings are detailed and easy to follow.

5) The authors share code, prompts, and data guidelines openly, which helps with reproducibility and future research.

**Weaknesses:**

1) The dataset size feels small compared to other multimodal benchmarks. About ten hours of content may not capture much variation in topics or speakers.

2) The evaluation relies mostly on automatic metrics. Some human checks or qualitative examples would make the results more convincing.

3) The discussion could go a bit deeper on why models fail.

**Questions:**

How consistent were the human annotations across different annotators and languages?

---

> ### Author Response · Authors · 2025-11-19
>
> We thank the Reviewer for their positive assessment and for recognizing the timeliness, careful design, and reproducibility of MCIF and the value of the extensive analyses. We address the raised concerns and questions below:
> # Weaknesses
> ## W1. Dataset size
> We acknowledge that MCIF (100 talks for summarization tasks, and 21 talks for the others) is smaller than large-scale multimodal benchmarks relying on synthetic data (as discussed in lines 078-081 and 145-149); however, its strength lies in **natural human-labeled and expert-vetted crosslingual design**, as synthetic data lacks the reliability and semantic richness of human-curated data. Each talk is fully human-annotated across four languages, three modalities, and 13 fine-grained tasks, resulting in **over 12,000 multimodal and crosslingual instances**, a scale comparable to or larger than prior human-annotated instruction-following benchmarks (e.g., SAKURA, Speech-IFEval). Moreover, our data selection process ensures, besides pure quality, high linguistic complexity (e.g., many named entities such as speakers’, models’, and datasets’ names), and speaker variation (e.g., mostly non native speakers, from different geographical areas). Lastly, we would like to point out that most test sets in speech tasks, such as ASR and ST, are about 2 hours (e.g., [LibriSpeech](https://ieeexplore.ieee.org/document/7178964), [FLEURS](https://ieeexplore.ieee.org/abstract/document/10023141), and [EuroParl-ST](https://www.mllp.upv.es/europarl-st/)). Overall, while we agree that increasing MCIF size would be beneficial (and possible, since we share all the details about the data creation process, including the guidelines), the high costs of human curation make this extension hardly feasible, and the current size is in line with many available benchmarks.
> ## W2. Evaluation metrics
> We rely on widely used (e.g., by the popular [Dynamic-SUPERB benchmark](https://openreview.net/forum?id=s7lzZpAW7T), and in the [IWSLT](https://aclanthology.org/2025.iwslt-1.44/) and [WMT](https://aclanthology.org/2024.wmt-1.1/) Evaluation Campaigns), reproducible metrics (WER, COMET, BERTScore) to systematically evaluate 23 models, which proved to be aligned with human judgment [[1](​​https://aclanthology.org/2025.wmt-1.24),[2](https://aclanthology.org/2025.iwslt-1.2/)]. Conducting large-scale human evaluation was *infeasible* given the number of models and languages, but **our paper does include manual checks and examples**: for instance, we manually inspected selected models’ outputs for recognition (lines 308-310), translation (lines 320-322), and short vs. long comparison (lines 386-393).
>
> ## W3. Discussion of model failures
> While a deeper analysis of why models fail was not the primary focus of this work, our benchmark enables such investigations. We provided a detailed analysis of failure cases, including long-form degradation (lines 382-304), sensitivity to prompt variations (lines 395-405), limited multimodal integration (Section 5.2), and sensitivity to QA type (Section 5.3). These observations reflect both current model limitations and architectural constraints, and are consistent across languages and modalities. Due to the 10-page limit (9 at submission), we prioritized presenting MCIF, its characteristics, and the analyses that illustrate the potential insights the benchmark can provide, along with key trends observed across the several evaluated models.
>
> # Question
> ## Q1. Annotation consistency
> All annotations were created by professional human annotators following detailed guidelines, available in our [**Anonymous GitHub**](https://anonymous.4open.science/r/mcif-1005/) repository. Cross-checks were performed by domain experts to maintain consistency across modalities and languages, ensuring a reliable and consistent ground truth for all evaluation tasks. All the details are reported in Appendix B.
>
> **In summary**, we thank the Reviewer for recognizing the timeliness, careful design, and reproducibility of MCIF. We addressed the concerns by clarifying that the dataset’s size reflects fully human and expert-vetted multimodal annotations, that our evaluation relies on widely adopted and reproducible metrics complemented by manual checks, and that our discussion already covers key failure patterns.

---

> > ### Comment · Reviewer_sQwM · 2025-11-21
> > **reply to authors**
> >
> > Thank you for your reply. To clarify, I think the discussion could go a bit deeper on why models fail (not where); current discussion in the paper is not enough (lacking examples, etc), and the information provided in 'analysis' is in the table already. About annotation consistency, can you provide the lines in Appendix where you discuss computation or evaluation of consistency?

---

> > > ### Author Response · Authors · 2025-11-24
> > >
> > > We thank the Reviewer for their reply. We are happy to discuss these two points:
> > > - Regarding the lack of examples, thanks to the Reviewer’s suggestion, we added them in Appendix F and integrated the pointers in our discussion in Section 5. In addition, we would like to clarify that Table 2 alone cannot provide the information discussed in the analysis part of the paper (Sections 5.2 and 5.3), as it shows the performance of the models by leveraging all the modalities and across the QA types. Instead, in Section 5.2, we analyze the performance of MLLMs by varying the modality to which they can access to (i.e., how the same model can effectively integrate various modalities), and, in Section 5.3, we break down the QA performance by category.
> > > - Regarding the consistency, it is important to note that MCIF annotations are English transcripts, questions, answers, and summaries as well as their translations in multiple languages. As these do not involve classification or labeling tasks, the computation of standard inter-annotator agreement metrics is neither applicable nor suitable. In our case, the primary objective is not to measure agreement between annotators, but to ensure the overall quality and reliability of the created ground-truth data. This quality was guaranteed by employing professionals for the creation of transcripts and translations, and by involving domain and task experts who are native speakers of the target languages to thoroughly review and validate their output. Several forms of consistency were also explicitly addressed:
> > >   - *Cross-lingual consistency* was ensured by design, as all original content (transcripts, questions, answers, and summaries) was first created in English and subsequently translated by professional translators.
> > >   - *Consistency with task guidelines* was maintained through expert verification, during which domain specialists not only assessed the linguistic quality of the professional translations, but also confirmed strict adherence to task guidelines and the correct use and translation of domain-specific terminology.
> > >   - *Cross-modality consistency* is ensured by definition, as the same ground-truth data are used for a given task across all modalities.
> > >
> > > Thanks to the Reviewer’s feedback, we further elaborated on this in Appendix B (lines 1177-1191) of the revised pdf.

---

> > > > ### Comment · Reviewer_sQwM · 2025-11-28
> > > > **Response to authors**
> > > >
> > > > Thanks for the update, I will keep my positive score.

---

### Official Review · Reviewer_Qo2z · 2025-10-31

**Soundness:** 3
**Presentation:** 3
**Contribution:** 2
**Rating:** 4
**Confidence:** 4

**Summary:**

This work introduces a new benchmark called MCIF, comprised of four classes of tasks (transcription, translation, question answering, and summarization) with aligned multimodal data (text, speech, and audio) from ACL lectures across four languages (English, German, Italian, and Chinese).

The benchmark uses human experts to provide annotations such as audio transcriptions and translations. Domain experts contribute the Q&A pairs, including unanswerable pairs.

The paper reports performance of 23 different models supporting various subsets of modalities (or all of them) and provides some analysis on prompt adherence and the contribution of individual modalities for those models that support multiple.

**Strengths:**

* The construction of the benchmark heavily relies on human experts, including domain experts of the material used, for ground truth annotation. It is really great to see such a rigorous collection protocol.
* The proposed task metrics (WER, BertScore, and COMET) are widely used and can be computed without relying on external APIs that may change over time.
* The paper presents results on their benchmark across 23 different models from various model families.

**Weaknesses:**

* The proposed benchmark is cross-lingual only from English into one of the three alternative languages (German, Italian, Chinese). Input audio and video (e.g. slides) are always English. This may limit the utility of the work for evaluating multi-lingual models’ capabilities in practice.
* While the work is clearly describing their contribution to be a benchmark about scientific talks, the data used seems quite narrow even within that domain; The benchmark exclusively uses recordings from presentations of main conference papers at ACL 2023. The limitation to this split was described as aiming to select videos newer than the knowledge cutoff of models to be evaluated, but this seems perhaps not well motivated as many of the evaluated models in the paper have a cutoff later than that (e.g. Phi 4, Gemma 3, GPT-oss, Gemini 2.5 Flash, for example). Selecting recordings from a single conference from a single year, and only main conference papers seems to significantly and perhaps arguably unnecessarily constrain the diversity of the produced benchmark (topics presented, recording setup and environment, speaker demographic,  etc. are presumably quite homogenous in the resulting work).
* The limited diversity of the benchmark, in my opinion, makes it a bit difficult to draw strong conclusions from the observed results. For example, section 5.1 argues that since no improvements were observed on the summarization task when adding speech or video in addition to the text transcript, this “underscores the limitations in multimodal integration”. However, an alternative interpretation may be that since the target summary is the abstract of the presented work, the content expected there may be particularly well captured in a transcript of the talk, since by nature such abstract would not include anything that may only occur on stage or any specifics of the visual presentation.
* The work focuses on models smaller than 20 billion parameters for its main study. While technical limitations are understandable, evaluating some stronger frontier models besides Gemini 2.5 Flash would be a welcome addition (GPT-5, Gemini 2.5 Pro, for example). Section 5.2 also does not seem to discuss Gemini 2.5 Flash, which is perhaps particularly limiting as the conclusion suggested, that “current MLLMs struggle to effectively integrate speech and video”, which may be particularly true for smaller model sizes, and notably Gemini 2.5 Flash was the best performing audio-video model reported in Table 2.
* The paper discusses instruction following through the lens of prompt adherence by comparing the results of $MCIF_{fix}$ and $MCIF_{mix}$. This seems a somewhat limited view of instruction following since the main signal will be whether the model will treat one (specific) prompt similar to 10 other semantically equivalent paraphrases. For example, assume a model that is very sensitive to the prompt - as long as the particular selected prompt for $M_{fix}$ achieves performance similar to the average of performances of $MCIF_{mix}$, this issue would not be visible in the chosen metrics. Additionally, the prompts themselves are relatively basic one-sentence instructions which may not exercise the instruction following capabilities of modern LLMs to their fullest extend.
* (Note that the first prompt for summarization in German as shown in Table 7 seems semantically slightly different to the other 9; it asks for a summary with at most 200 words instead of approximately 200 words; I am unable to verify all prompts, but minor semantic differences may also explain much of the observed fluctuation in scores?)

**Questions:**

* In table 1, the context type for the text-inputs is “-“. Does that mean that for the short context setting we still provide the full transcript and only non-text modalities are shortened to the relevant region of interest? If not, why is the performance of text input for short context not reported for translation and QA?
* In the main results in table 2, are all question pairs used for QA regardless of whether the questions can be answered with the provided input modalities? If so, what is the “correct” way to answer questions that are not answerable with the provided input modalities (including questions that are generally unanswerable)?

---

> ### Author Response · Authors · 2025-11-19
>
> We thank the Reviewer for their detailed feedback and for recognizing the rigor of our data collection, annotation protocol, and large-scale model evaluation. Below, we address each main concern and clarify several design motivations.
>
> # Weaknesses
>
> ## W1. Scope of cross-lingual coverage
> We acknowledge that extending the scope of languages, both on the source side (where we have only English) and on the target side (where we have 4 languages), would improve the resource. However, on the one hand, scientific talks are overwhelmingly produced in English, as it serves as a *lingua franca* in scientific communication, thus limiting the availability of multimodal material in other languages for this specific domain. On the other hand, extending the target language coverage entails human effort and costs that go beyond our current possibilities. Still, we cover 4 languages from 4 different families. For this reason, we believe that this valuable extension does not invalidate our contribution and findings and can be left to future work.
>
> ## W2. and W3. Data Diversity
> We respectfully disagree with the Reviewer’s assessment that the benchmark is overly narrow or homogeneous. Although ACL mostly focuses on NLP, it covers a broad range of AI-related topics, spanning from multimodality to explainability and ethics. Moreover, the recording setup is far from uniform: each presenter recorded their talk independently, using their own equipment and environment, resulting in significant variation in audio and video quality, background, and presentation style. The speaker demographic is equally diverse, as ACL is an international conference that features researchers from around the world. According to [the official ACL 2023 report](https://aclanthology.org/2023.acl-long.report.pdf), participants and authors represent a wide range of nationalities from Europe, the Americas, Asia, and beyond, which is reflected in our dataset as well. To better highlight these characteristics, we revised the abstract and Section 3.1 accordingly. Regarding the selection of ACL 2023 videos, we chose this edition because it represents the most recent year for which full video material is publicly available through the ACL Anthology. These talks were released during 2024, meaning their content postdates the training data of most evaluated models. Hence, our choice minimizes potential data contamination, to the best of our ability, rather than constraining diversity. Additionally, we disagree with the suggestion that strong LLM performance on summarization can be attributed to the abstract being *“particularly well captured”* in transcripts. Transcripts are derived directly from the speech and convey the same content; in fact, speech provides richer semantic cues than text alone. Rather than reflecting an artifact of dataset design, the fact that text-only LLMs improve on summarization tasks compared to SpeechLLMs reinforces our key finding: current multimodal models struggle to integrate speech and visual information effectively, as extensively discussed in Section 5.2.
>
> ## W4. Model scale and inclusion of larger systems
> We acknowledge the value of including stronger proprietary models (e.g., Gemini 2.5 Pro, GPT-5) to establish an upper performance bound; we do include a closed (and presumably very large) system, Gemini 2.5 Flash. While budget constraints prevented adding more proprietary models, our evaluation spans systems from 3B to 20B parameters, covering a wide range of **accessible and freely available** models. Furthermore, many proprietary systems do not consistently provide full multimodal APIs (e.g., GPT-5 does not directly support videos with embedded audio processing, at the moment, as it calls the Whisper APIs to provide the transcription first) or reproducible inference access, whereas our work ensures a fully reproducible evaluation framework alongside the dataset.
>
> ## W5. Prompt Variation
> We agree that our evaluation captures only a subset of instruction-following behavior, specifically robustness to semantically equivalent prompts in realistic, task-driven contexts. At the same time, we note that using a simple one-sentence instruction reflects common practice for prompting most SpeechLLMs, VideoLLMs, and MLLMs, and mirrors the type of instructions these models were trained on. While more sophisticated techniques (such as prompt engineering, or chain-of-thought reasoning) could improve individual model performance, the core goal of instruction following is for models to correctly respond to any reasonable instruction (not only to carefully engineered ones). Our design, therefore, ensures fair and comparable evaluation across all models without introducing confounding factors. Exploring the impact of more complex prompting strategies is an important direction for future work, and MCIF provides a foundation for such studies. We thank the Reviewer for the suggestion, and we included this discussion in the Conclusions of our work.

---

> > ### Author Response · Authors · 2025-11-19
> >
> > ## WP6. Issue in German prompt
> > We agree with the Reviewer that the initial German prompt did not convey the same meaning as its English counterpart. Originally, all prompts were formulated in a similar way to the first German one, following the constraints of *ACL conferences (requiring up to 200 words for the abstract), but we later revised them as we noticed that this constraint was not actually strictly respected. In revising them, we inadvertently missed updating this single instance. We have now corrected it to  *“Fasse den englischen Inhalt in einem Abstract mit ungefähr 200 Wörtern zusammen.”* and re-ran all models for the German summarization task. The updated summarization results for English to German are reported in the revised version of the paper.
> >
> > # Questions
> > ## Q1. Observation on textual modality in Table 1
> > We thank the Reviewer for noticing this small inconsistency in our table. The context should be *LONG* instead of “-” as we don’t have sentence-level textual segmentation aligned with short-form speech and video. We have corrected it in Table 1 of the revised pdf.
> >
> > ## Q2. QA pairs considered in Table 2 results
> > All QA pairs were included in the overall results reported in Table 2, while a detailed breakdown by question category highlights that NA answers were consistently poor across models. Excluding any subset (e.g., audio-only questions) would have broken the benchmark’s parallel design of MCIF; therefore, all questions are retained to preserve systematic comparability across modalities and languages.
> >
> > **In summary**, we thank the Reviewer for their insightful comments and questions, which helped us to improve the paper. We believe that our responses addressed the main concerns raised in the review, highlighting that MCIF is both realistic and diverse (also considering the current data availability), and that its design supports the evaluation of instruction-following, multimodal integration, and prompt robustness under controlled semantic perturbations.

---

### Official Review · Reviewer_MiXV · 2025-10-31

**Soundness:** 3
**Presentation:** 3
**Contribution:** 3
**Rating:** 6
**Confidence:** 4

**Summary:**

This paper presents MCIF, the first human-annotated, multimodal, crosslingual instruction-following benchmark built from scientific presentations. It evaluates large multimodal language models (MLLMs) across three modalities (text, speech, and video), four languages (English, German, Italian, and Chinese), and 13 tasks grouped into four macro categories: recognition, translation, question answering (QA), and summarization. The dataset includes short-form and long-form contexts, enabling the study of model performance across input complexity.

**Strengths:**

1. The paper tackles a critical gap in evaluating instruction-following across multiple languages and modalities - an under-explored but essential part of MLLMs.

2. MCIF is carefully constructed with parallel multimodal and multilingual alignment, allowing systematic and fair cross-dimensional comparisons—something absent in prior benchmarks. Using human-curated data from scientific talks ensures higher linguistic and contextual richness than synthetic or crowdsourced datasets. The academic domain provides realistic, challenging material.

3. It conducts comprehensive evaluation over 23 diverse models (open and closed-source, text-only and multimodal) under both short- and long-form settings provides broad insight into model generalization and limitations. And it also introduces MCIF_fix vs. MCIF_mix is a thoughtful addition—it measures how sensitive models are to prompt variations, reflecting real-world user interaction diversity.

**Weaknesses:**

1. Scale issue: despite being well-curated, MCIF covers only ~10 hours of content and 21 talks, which may limit statistical robustness and task diversity.

2. Human baseline: it is better to establish a human baseline, making it clear how far current systems are from “human-level” multimodal understanding.

3. Evaluation metrics: Automatic evaluation on WER, COMET, and BERTScore may not well capture multimodal grounding or factual consistency - especially for tasks like question answering or summarization.

**Questions:**

1. How you control the quality to ensure annotation consistency across languages and modalities, especially for translations and Q&A?

2. Can you elaborate on whether degradation in long-form tasks is primarily due to model architecture (context window limits) or instruction misalignment? or any other potential reasons?

3. For the variation shown between MCIF_fix and MCIF_mix, do you suggest any strategies to improve real-world prompt robustness?

---

> ### Author Response · Authors · 2025-11-19
>
> We thank the Reviewer for their assessment, recognizing the novelty and careful design of MCIF, as well as its comprehensive evaluation. We address the Reviewer’s main concerns and questions below.
>
> # Weaknesses
> ## W1. Dataset scale
> We acknowledge that MCIF (100 talks for summarization tasks, and 21 talks for the others) is smaller than large-scale synthetic datasets; however, its strength lies in **natural human-labeled and expert-vetted crosslingual design**, not raw duration. Each talk is fully human-annotated across four languages, three modalities, and 13 fine-grained tasks, resulting in **over 12,000 multimodal and crosslingual instances**, a scale comparable to or larger than prior human-annotated instruction-following benchmarks (e.g., SAKURA, Speech-IFEval). Moreover, our data selection process ensures, besides pure quality, high linguistic complexity (e.g., many named entities such as speakers’, models’, and datasets’ names), and speaker variation (e.g., mostly non native speakers, from different geographical areas). Lastly, we would like to point out that most test sets in speech tasks, such as ASR and ST, are about 2 hours (e.g., [LibriSpeech](https://ieeexplore.ieee.org/document/7178964), [FLEURS](https://ieeexplore.ieee.org/abstract/document/10023141), and [EuroParl-ST](https://www.mllp.upv.es/europarl-st/)). Overall, while we agree that increasing MCIF size would be beneficial (and possible, since we share all the details about the data creation process, including the guidelines), the high costs of human curation make this extension hardly feasible, and the current size is in line with many available benchmarks.
>
> ## W2. Human baselines
> We agree human baselines would be informative, but conducting human evaluation for **23 models × 13 tasks × 4 languages** entails a cost that we cannot support. Instead, our focus is on a broad, reproducible model comparison, for which we are going to release the complete evaluation framework.
>
> ## W3. Evaluation metrics reliability
> We understand the Reviewer’s concern about potential limitations of automatic metrics. However, we would like to point out that *i)* the Findings of both the [IWSLT 2025 Shared Tasks](https://aclanthology.org/2025.iwslt-1.44/) and [WMT25 Metric Shared Task](​​https://aclanthology.org/2025.wmt-1.24) underlined that automatic (WER, COMET, BERTScore) and human evaluation were completely in agreement for tasks such as recognition and translation, and *ii)* for more complex tasks such as summarization (for which we adopted BERTScore), both prior [[1](https://aclanthology.org/2021.alta-1.9/),[2](https://aclanthology.org/D19-5817.pdf)] and recent [[3](https://aclanthology.org/2025.iwslt-1.2/)] papers demonstrated a very high correlation between BERTScore and human judgement, making it the most suitable metrics for these tasks. We agree that this aspect opens space for developing better multimodal evaluation metrics, including multimodal grounding as the Reviewer has pointed out, and we view MCIF as a valuable evaluation framework to build upon in future work.

---

> > ### Author Response · Authors · 2025-11-19
> >
> > # Questions
> > ## Q1. Annotation quality and consistency
> > As mentioned in *Appendix B*, all annotations were **performed and cross-reviewed by bilingual experts** for each language pair and task, following the guidelines available in the [**Anonymous GitHub**](https://anonymous.4open.science/r/mcif-1005/) provided at review time. For transcripts and translations (including the summaries), the annotators started from automatic outputs and they post-edited them. For QA, the pairs were completely human-created following the criteria in the guidelines: consistency is always ensured among languages since the QA pairs were created in English and then translated (and double-checked) into the three other languages. Lastly, all outputs were double-checked by the domain experts using the same guidelines. Also, the consistency across modalities is ensured by definition, as we use the same annotations for the same tasks across different modalities.
> >
> > ## Q2. On long-form degradation causes
> > The observed drop in long-form performance primarily stems from the fact that most models are not trained on extended contexts. While the feature concatenation strategy of some of them (e.g., Ola) enables processing longer sequences, it also exposes architectural limitations in maintaining contextual coherence and generating complete outputs. This remains a key challenge, especially for SpeechLLMs and MLLMs, as we discussed in the SHORT vs LONG paragraph (lines 382-394) and Section 5.2. Moreover, we do not see any potential for instruction misalignment as the instructions are the same across the long and short settings.
> >
> > ## Q3. Improving prompt robustness
> > We thank the Reviewer for highlighting this. Our results on $MCIF_{fix}$ vs. $MCIF_{mix}$ suggest that *robustness to prompt variation* is an open challenge. Strategies that may mitigate this include prompt-embedding alignment, instruction tuning on paraphrased data, and contrastive training between equivalent instructions. We have expanded this discussion in Section 6 of the revised pdf.
> >
> > **In summary**, we appreciate the Reviewer’s recognition of MCIF’s novelty and comprehensive design. The benchmark’s human-curated, crosslingual, and multimodal nature fills a crucial gap in evaluating instruction-following models, and we incorporated the Reviewer’s feedback in Section 6 to further strengthen the paper.

---

### Meta-Review · Area_Chair_xquA · 2026-01-09

**Summary:**

This paper presents MCIF, the first human-annotated, multimodal, crosslingual instruction-following benchmark built from scientific presentations. It received scores of 4668. On the positive side, reviewers commented that the benchmark is carefully constructed, and the authors conduct comprehensive evaluation over 23 diverse models. On the other hand, reviewers also showed concerns that (1) the diversity of the benchmark seems limited, and (2) the benchmark only covered a somewhat limited view of instruction following, which the AC fully agrees. The authors share code, prompts, and data guidelines openly. On balance, the AC would like to recommend acceptance by the end, and let the research community to judge the longer-term impact of the benchmark.

**Reviewer Concerns:**

Concerns adequately addressed:

1. The benchmark is in small scale.

2. How you control the benchmark quality.

3. Evaluation metrics reliability.


Concerns insufficiently addressed:

1. The paper discusses instruction following through the lens of prompt adherence by comparing results of  MCIF_fix and MCIF_mix. This seems a somewhat limited view of instruction following. Additionally, the prompts themselves are relatively basic one-sentence instructions which may not exercise the instruction following capabilities of modern LLMs to their fullest extend. The AC fully agrees on this.

2. While the work is clearly describing their contribution to be a benchmark about scientific talks, the data used seems quite narrow even within that domain. The limited diversity of the benchmark makes it a bit difficult to draw strong conclusions from the observed results.

3. Evaluating some stronger frontier models besides Gemini 2.5 Flash would be a welcome addition.

**Reviewer Scores:**

Overall, the paper received good average scores. However, as several concerns remain insufficiently addressed, it is unlikely that reviewers would be inclined to increase their scores further.

---

### Decision · Program_Chairs · 2026-01-26

Accept (Poster)